# Robust and Efficient Fine-tuning of LLMs with Bayesian Reparameterization of Low-Rank Adaptation

**Ayan Sengupta**[*,◇]                                          *ayan.sengupta@ee.iitd.ac.in*
*Indian Institute of Technology Delhi, India*

**Vaibhav Seth**[*]                                          *Vaibhav.Seth.mt121@maths.iitd.ac.in*
*Indian Institute of Technology Delhi, India*

**Arinjay Pathak**[*]                                          *aiz248678@scai.iitd.ac.in*
*Indian Institute of Technology Delhi, India*

**Aastha Verma**                                          *cs1221607@iitd.ac.in*
*Indian Institute of Technology Delhi, India*

**Natraj Raman**                                          *natraj.raman@jpmorgan.com*
*JPMorgan AI Research*

**Sriram Gopalakrishnan**                                          *sriram.gopalakrishnan@jpmchase.com*
*JPMorgan AI Research*

**Niladri Chatterjee**                                          *niladri@maths.iitd.ac.in*
*Indian Institute of Technology Delhi, India*

**Tanmoy Chakraborty**[◇]                                          *tanchak@iitd.ac.in*
*Indian Institute of Technology Delhi, India*

**Reviewed on OpenReview:** *https://openreview.net/forum?id=2HFmicB8kh*

## Abstract

Large Language Models (LLMs) are highly resource-intensive to fine-tune due to their enormous size. While low-rank adaptation is a prominent parameter-efficient fine-tuning approach, it suffers from sensitivity to hyperparameter choices, leading to instability in model performance on fine-tuning downstream tasks. This paper highlights the importance of effective parameterization in low-rank fine-tuning to reduce estimator variance and enhance the stability of final model outputs. We propose `MonteCLoRA`, an efficient fine-tuning technique that employs Monte Carlo estimation to learn an unbiased posterior estimation of low-rank parameters with low expected variance, stabilizing fine-tuned LLMs with only $\mathcal{O}(r)$ additional parameters, for a given rank $r$. `MonteCLoRA` shows 0.5% and 1.6% improvements in accuracy and robustness over unregularized low-rank adaptation method on natural language understanding tasks with pre-trained RoBERTa-base. Furthermore, in generative tasks with pre-trained LLaMA-1-7B and LLaMA-3.2-3B-Instruct, `MonteCLoRA` demonstrates robust performance with 50% and 62% lower spreads respectively than the contemporary efficient fine-tuning methods. The theoretical and empirical results presented in the paper underscore how parameterization and hyperpriors balance exploration-exploitation in the low-rank parametric space, therefore leading to more optimal and robust parameter estimation during efficient fine-tuning.

---

[*]Equal contribution
[◇]Corresponding author

# 1 Introduction

The rise of large language models (LLMs) has initiated a transformative shift in natural language processing, revolutionizing an extensive array of tasks (Zhao et al., 2023; Chang et al., 2024). Vaswani et al. (2017) introduced the self-attention-based Transformer architecture, which is capable of more efficient handling of long-range dependencies in texts than prior methods that relied on recurrent neural networks (RNNs) and convolutional neural networks (CNNs). Since then, a monumental shift has begun in developing Transformer-based pre-trained language models (PLMs) for solving a wide range of tasks involving natural languages. Over the past few years, the size of these PLMs has dramatically increased from multi-million parameter BERT (Devlin et al., 2018), RoBERTa (Liu et al., 2019), T5 (Raffel et al., 2020) models to recently developed multi-billion parameter DeepSeek (Liu et al., 2024a), LLaMA (Grattafiori et al., 2024b; Touvron et al., 2023), Falcon (Almazrouei et al., 2023) and Mistral (Jiang et al., 2023) models. Scaling laws of language models (Kaplan et al., 2020) suggest that the superior performance of these models scales with pre-training data size and the required computation. With deeper and larger models and more extensive pre-training, these models exhibit emerging properties such as zero-shot and few-shot in-context learning (Brown et al., 2020), complex reasoning, and generalization capabilities. Despite these emerging properties, LLMs require fine-tuning on downstream tasks for competitive performance (Liu et al., 2022b) and domain and task adaptation.

Given their enormous size and computational requirements, fine-tuning LLMs on every downstream task is often unrealistic and computationally infeasible. For feasibly fine-tuning LLMs, parameter-efficient fine-tuning (PEFT) techniques such as Adapters (Houlsby et al., 2019), selective fine-tuning (Zaken et al., 2021), and low-rank adaptation (Hu et al., 2022) have become immensely popular. Among these PEFT methods, low-rank adaptation (LoRA) has garnered significant attention due to its flexibility, adaptiveness, and ability to mitigate catastrophic forgetting during fine-tuning. LoRA reparameterizes pre-trained model weights to a lower dimension, such that only the low-rank matrices are tuned during fine-tuning, keeping the weights of the original pre-trained model frozen. The low-rank decomposition significantly reduces the number of trainable parameters during fine-tuning, offering great computational benefits. For instance, with a latent rank of 8, the number of trainable parameters of a RoBERTa-base (Liu et al., 2019) model can be reduced by 99% (from 110M to 0.3M) with LoRA. Despite its effectiveness, recent studies (Liu et al., 2022c; Valipour et al., 2022; Biderman et al., 2024) showed that LoRA is sensitive to hyperparameters like learning rate and training batch size and often requires longer training iterations for convergence.. Figure 1 illustrates the performance of full fine-tuning (where we fine-tune the whole pre-trained LLM) and LoRA fine-tuning on different natural language understanding tasks with the pre-trained RoBERTa-base model under different learning rates and training batch sizes. The results highlight that the distribution spread for accuracy (difference between maximum and minimum accuracy) on the validation dataset can go up to 17 and 24 points for LoRA and full fine-tuning, respectively. Although the inter-quartile range (IQR) for LoRA remains modest for most tasks, the high distribution spreads suggest that both LoRA and full fine-tuning are sensitive to marginal cases. However, as we discuss in the subsequent sections, distributional spread is not always the best measure of robustness due to outlier sensitivity (full fine-tuning more sensitive to outliers than LoRA) and distribution median forms a better metric for measuring robustness. We observe that median performance of full fine-tuning is higher than LoRA in 4 out of 5 tasks. We further observe that the accuracy spread for LoRA, particularly for generative tasks remains considerably high relative to alternative methods and median accuracy is also fairly low compared to other methods, underscoring the need for more stable formulation of low-rank adaption methods for fine-tuning LLMs. These observations highlight that careful consideration of hyperparameter selection is necessary to balance adapting new knowledge and preserving the pre-trained knowledge. The most obvious way to figure out the most appropriate hyperparameters is to perform extensive hyperparameter tuning (Tribes et al., 2023; Xie et al., 2024). However, unlike small-scale machine learning models, performing grid or random search within the hyperparameter space of LLMs is very costly and impractical. Bayesian methods (Wilson & Izmailov, 2020; Wang & Yeung, 2020), on the other hand, offer an organized solution to hyperparameter sensitivity by marginalizing the predictive distribution. Through appropriate knowledge priors, Bayesian methods diminish the importance of hyperparameter tuning (Papamarkou et al., 2024a) and offer robust alternatives to post-hoc regularization techniques while training on small datasets.

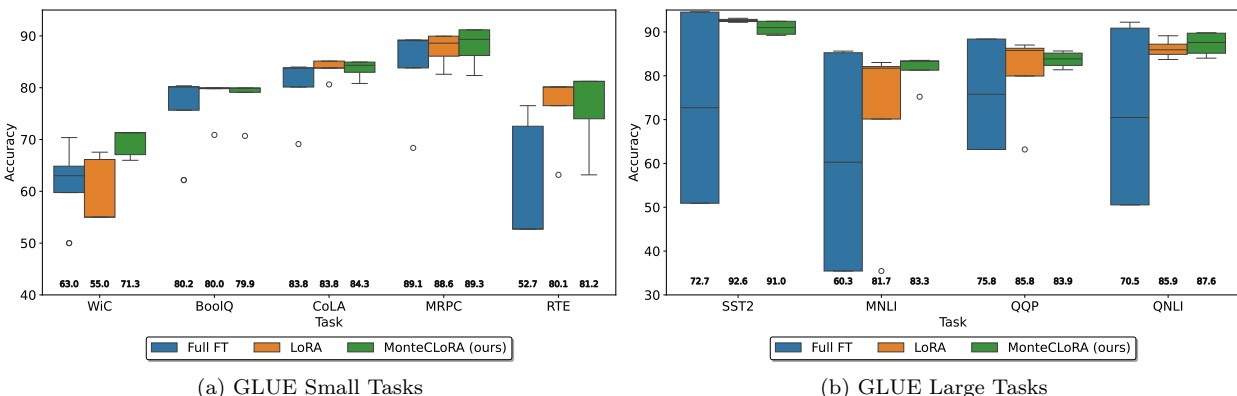

(a) GLUE Small Tasks

(b) GLUE Large Tasks

Figure 1: The distribution of validation accuracy for the fine-tuned RoBERTa-base under different learning rates and batch sizes. We present the findings for two approaches: full fine-tuning (denoted as Full FT) and low-rank adaptation (denoted as LoRA). Full FT tends to be more sensitive to outliers and exhibits a wider spread (the difference between the maximum and minimum scores) compared to LoRA. However, LoRA's median performance is lower than that of Full FT in 4 out of 5 small GLUE tasks, suggesting that Full FT offers greater robustness than LoRA when the fine-tuning data is limited. In the larger GLUE tasks, LoRA gives a better median value but the spread of accuracy is much greater for Full FT. Our proposed method, `MonteCLoRA` demonstrates better average spread and median performance than both Full FT and LoRA. This highlights the significance of appropriate parameterization for achieving stable low-rank adaptation in LLMs.

Motivated by the advantages of Bayesian methodologies, we propose a robust low-rank adaptation method for fine-tuning LLMs efficiently. Contrary to the existing sensitivity studies of LoRA, our work provides a structured overview of the challenges faced by LoRA and the full fine-tuning method and proposes a systematic approach to mitigate these challenges. To overcome the sensitivity challenges of LoRA fine-tuning, we propose a Monte Carlo-enhanced low-rank adaptation method, `MonteCLoRA`, which learns posterior distributions of low-rank parameters with appropriate prior distributions. `MonteCLoRA` parameterizes the low-rank parameters as a mixture of multivariate Gaussian distributions, where each distribution's precision matrix is assumed to follow Wishart knowledge prior. Through Monte Carlo estimation from multiple parameters sampled from the parameter space, `MonteCLoRA` stabilizes the reparametrized parameters and generates robust and unbiased low-rank adaptation for LLMs.

Our theoretical and empirical results justify the importance of parameterization for low-rank adaptation of LLMs. We perform thorough empirical analysis with five natural language understanding (NLU) and six natural language generation (NLG) tasks with three pre-trained LLMs — RoBERTa-base (Liu et al., 2019), LLaMA-1-7B (Touvron et al., 2023), abd LLaMA-3.2-3B-Instruct (Grattafiori et al., 2024a). Empirical results on NLU tasks suggest that `MonteCLoRA` is more stable, where the average spread of accuracy distribution is 10% lower than LoRA and 50% lower than full fine-tuning. In terms of the robustness metrics, `MonteCLoRA` is 5% more robust and achieves 2% better accuracy than LoRA fine-tuning. Remarkably, on Commonsense NLG tasks with LLaMA-1-7B, `MonteCLoRA` has a 53% lower spread (2.19 points) than LoRA (4.69 points) with zero-shot validation accuracy distribution. With LLaMA-3.2-3B-Instruct `MonteCLoRA` reduces the spread by 62% on average and exhibits 13.3% higher robustness over LoRA. Our further in-depth analysis highlights the superiority of `MonteCLoRA` in terms of stable and faster convergence while fine-tuning LLMs.

The key contributions of our work can be summarized as follows[1]

---

[1]The source code of `MonteCLoRA` is made available at `https://github.com/LCS2-IIITD/MonteCLoRA`.

- Our work provides an in-depth theoretical analysis of the impact of hyperparameters on fine-tuning LLMs and highlights the key challenges with low-rank adaptation techniques. To the best of our knowledge, no comprehensive study exists on the sensitivity analysis of LLM fine-tuning.

- We propose a Bayesian alternative to the low-rank adaptation of LLMs for estimating trainable parameters during fine-tuning. The paper theoretically justifies the robustness of the posterior estimation.

- The proposed fine-tuning method adds only $\mathcal{O}(r)$ additional parameters for posterior estimation and is shown to be effective in both the performance and the stability of the fine-tuned LLMs.

- We provide a thorough empirical study where we demonstrate the effectiveness of the proposed fine-tuning methods with two pre-trained language models – RoBERTa-base and LLaMA-1-7B on five NLU and six NLG (commonsense reasoning) tasks.

The paper is organized as follows. Section 2 describes the related work on efficient fine-tuning strategies for LLMs. Section 3 elaborates on the background concepts used in the paper. Section 4 describes our proposed method, `MonteCLoRA`, along with the theoretical results obtained in this work. Section 5 describes the experimental details used in the empirical study done in the paper. In Section 6, we present the results of the empirical study. Section 7 highlights a few case studies to analyze robust fine-tuning of LLMs. Additional background materials and supplementary results are furnished in the appendix.

## 2 Related Work

In this section we briefly describe the related work around robust fine-tuning of LLMs. We divide this section into three broad subjects – (1) fine-tuning methods for LLMs, (2) efficient fine-tuning methods, and (3) Bayesian methods for robust fine-tuning.

**Fine-tuning LLMs.** Fine-tuning refers to continually training PLMs on a smaller, task-specific dataset. Fine-tuning allows the model to adapt its generalized capabilities to perform better on particular tasks, domains or applications that are not prevalent during the pre-training phase. For the given task-specific training data $Z = \{(x_i, y_i)\}$, in full fine-tuning of LLMs, the model starts with pre-trained weights $\theta_0$ and updates to $\theta = \theta_0 + \Delta\theta$ by maximizing the conditional language modeling objective

$$\max_\theta \sum_{(x,y)\in Z} \sum_{t=1}^{|y|} \log P_\theta(y_t \mid x, y_{<t}).$$

In one of the foundation works, Devlin et al. (2018) demonstrated that models pre-trained on large text corpora could be fine-tuned with just one additional output layer to perform a wide range of tasks, effectively transferring learned knowledge to specific tasks. Howard & Ruder (2018) introduced the concept of universal model fine-tuning, emphasizing the importance of discriminative fine-tuning and gradual unfreezing for each model layer to adapt effectively to different downstream tasks. This approach helps mitigate the catastrophic forgetting problem often observed in LLMs. However, as the models became larger and more capable, fully fine-tuning LLMs turned out to be more computationally infeasible.

**Parameter-efficient methods of fine-tuning.** PEFT aims to adapt large PLMs to specific tasks by optimizing a small subset of the model's parameters rather than undertaking the computationally expensive training of the entire pre-trained model. These techniques significantly reduce the computational overhead and memory usage typically associated with fine-tuning large models. There are three broad classes of PEFT techniques – additive, selective, and reparameterization-based. Additive PEFT strategies (Houlsby et al., 2019; Pfeiffer et al., 2020) modify the underlying model's architecture by injecting new additive trainable parameters. Selective PEFT methods (Zaken et al., 2021; Sung et al., 2021) select a subset of parameters for fine-tuning, and the rest remain frozen. On the other hand, reparameterization-based PEFT methods introduce low-dimensional reparameterized parameters for fine-tuning.

One of the most prominent efficient fine-tuning methods, LoRA (Hu et al., 2022), uses low-rank reparameterization to reduce the learnable parameter space to a lower-dimensional space. Subsequently, several

alternatives to LoRA have been proposed, mostly keeping efficiency and downstream performance in focus. One such method, AdaLoRA (Zhang et al., 2023), dynamically allocates parameter budgets among weight matrices based on their importance, employing singular value decomposition to optimize incremental updates. Sparse LoRA (SoRA) extends LoRA by incorporating a gate unit optimized with a proximal gradient method (Ding et al., 2023), allowing dynamic adjustments to the intrinsic rank during training and effectively controlling the sparsity of updates. DoRA (Liu et al., 2024b) decomposes the pre-trained weight into magnitude and direction components for fine-tuning, utilizing LoRA for efficient directional updates, which enhances both the learning capacity and training stability without increasing inference overhead. Although these PEFT methods demonstrate superior downstream performance in terms of extrinsic metrics like accuracy, these methods are susceptible to task-specific configurations. For instance, SoRA depends on tuning the sparsity controls, introducing instability during training due to abrupt changes in sparsity levels. Moreover, these methods often overfit when fine-tuned on small datasets and generate overconfident predictions during inference (Yang et al., 2024).

**Bayesian Methods for Model Robustness.** Bayesian frameworks are particularly effective in handling model uncertainty (Daxberger et al., 2021; Zhang et al., 2021; Deng et al., 2022), offering a robust solution to this issue by enabling a probabilistic interpretation of model weights, which helps in assessing the uncertainty of the predictions. Papamarkou et al. (2024b) strongly argued that Bayesian deep learning is more favorable over similar frequentist alternatives. This is because Bayesian methods provide advantages that can help overcome many of the challenges that deep learning faces. Bayesian deep learning is known to reduce the importance of hyperparameter tuning by incorporating relevant hyper-priors. Bayesian deep learning is also known to enable domain knowledge priors, as opposed to post-hoc regularization on small datasets. Uncertainty quantification is another aspect where Bayesian methods gain an advantage over the alternative frequentist methods by using it to improve the reliability of the decision-making process, which helps the model generalize better on out-of-distribution inputs. Also, by dynamically updating prior beliefs in response to new evidence, Bayesian frameworks allow selective retention of valuable information from previous tasks while adapting to new ones. This mitigates the issue of model decay, which occurs in static models, assuming that underlying data patterns remain constant over time. In a recent development, Laplace-LoRA (Yang et al., 2024), a Bayesian post-hoc treatment of LoRA was introduced that leverages Laplace approximation to estimate the posterior distributions of the parameters involved in the low-rank adaptations, significantly improving the calibration of fine-tuned models. Albeit a robust solution to the overconfidence problem of PEFT methods, Laplace-LoRA is a post-hoc calibration method requiring longer training iterations to bring the low-rank parameters from an unstable basin (a subspace associated with the same local optimum) to a more stable parametric space. Therefore, Laplace-LoRA often leads to sub-optimal downstream performance.

**Uniqueness of `MonteCLoRA`.** Our method balances performance and stability through Monte Carlo estimation over the low-dimensional parametric space. With appropriate parametric assumptions, `MonteCLoRA` can provide an unbiased and robust estimate for the posterior distribution, providing excellent stability and performance gain. Moreover, unlike existing Bayesian methods like Laplace-LoRA, `MonteCLoRA` can be used on both ad-hoc and post-hoc basis and, therefore, can follow the same optimization path as the other PEFT methods to learn the reparameterized model parameters, ensuring better flexibility and adaptiveness.

## 3 Background

In this section we elaborate on the concepts of LoRA for fine-tuning LLMs. We also lay down the theoretical motivation behind robust low-rank parameterization. All the mathematical notations used in this section are summarized in Table 1.

### 3.1 LoRA for Fine-tuning LLMs

LoRA leverages low-rank matrix decomposition to incorporate trainable parameters that adapt the pre-trained model's weights in a parameter-efficient manner. Consider a pre-trained weight matrix $W_0 \in \mathbb{R}^{n_{\text{out}} \times n_{\text{in}}}$. LoRA constrains its update by expressing it through a low-rank decomposition as follows:

$$W = W_0 + \Delta W = W_0 + BA,$$

| Notation Type | Notation | Description |
|---|---|---|
| Low-rank adaptation | $W_0$ | Pre-trained weight matrix |
| | $n_{in}$ | Input dimensions of $W_0$ |
| | $n_{out}$ | Output dimensions of $W_0$ |
| | $r$ | LoRA rank |
| | $A$ | LoRA A matrix |
| | $B$ | LoRA B matrix |
| Optimization | $J$ | Objective or loss function |
| | $\theta$ | Model parameters |
| | $\nabla J$ | Gradient of the loss function |
| | $L$ | Lipschitz constant of the loss function |
| | $\sigma$ | Standard deviation bound of the stochastic gradients |
| | $\rho$ | Stochastic noise in the gradient |
| | $H$ | Hessian matrix of the loss function |
| | $\lambda$ | Eigenvalue of a matrix |
| | $\xi$ | Point for mean value theorem |
| | $\gamma$ | Gaussian random variable added to the model parameter |
| | $\Lambda_{max}$ | Maximum eigenvalue of Hessian |

Table 1: Glossary of all the mathematical notations used in Section 3.

where $B \in \mathbb{R}^{n_{out} \times r}$, $A \in \mathbb{R}^{r \times n_{in}}$, and the rank $r \ll \min(n_{in}, n_{out})$. During training, $W_0$ remains frozen and is not subject to gradient updates, while the matrices, $A$ and $B$, contain the trainable parameters. Both $W_0$ and $\Delta W = BA$ are applied to the same input, and their resulting output vectors are summed element-wise. This approach allows LoRA to efficiently adapt large pre-trained models to new tasks with a relatively small number of additional parameters. Figure 2 provides a pictorial illustration of LoRA.

## 3.2 Sensitivity of Gradient Descent to Hyperparameters

Stochastic gradient descent (SGD) is a widely popular optimization technique for training neural networks. In SGD, we calculate the gradient of the training objective function with respect to the learnable neural parameters on a sample training batch and iteratively update the parameters towards the edge of a basin of optima, as shown by Izmailov et al. (2019), where the training objective is minimized. In this section, we describe some of the properties of SGD to understand how different hyperparameters such as optimizer learning rate impacts the optimization process.

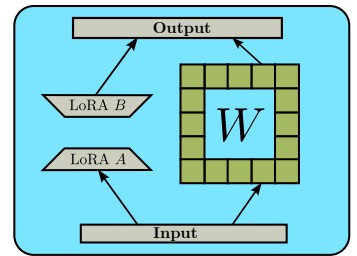

Figure 2: Illustration of the LoRA fine-tuning framework.

### 3.2.1 Factors Influencing Convergence of SGD

The convergence of SGD is known to depend on the Lipschitz constant through its influence on the choice of step sizes. Several studies have established an inverse relationship between the maximum step sizes that ensure convergence and the learning rate. Ghadimi & Lan (2013), in their seminal work, demonstrated that selecting the learning rate within a specific range guarantees the convergence of the SGD algorithm, based on foundational assumptions in stochastic optimization. The following lemmas formalize the relationships between the convergence of SGD with the step size of the algorithm[2].

**Lemma 3.1** *Let $J : \mathbb{R}^n \to \mathbb{R}$ be a twice continuously differentiable function, where $n \in \mathbb{N}$ is the dimension the space of model parameters. Then the gradient $\nabla J$ is Lipschitz continuous with Lipschitz constant $L$ equal to the supremum of the maximum eigenvalue of its Hessian matrix $H(\theta) = \nabla^2 J(\theta)$*

$$L = \sup_{\theta \in \mathbb{R}^n} \lambda_{max}(H(\theta)). \tag{1}$$

---

[2]Proofs of Lemma 3.1 and 3.2 are furnished in Section A of Appendix.

**Lemma 3.2** *Let $J(\boldsymbol{\theta})$ be the loss function over trainable model parameters $\boldsymbol{\theta} \in \mathbb{R}^n$, and $\boldsymbol{\gamma}$ represents Gaussian noise with zero mean. We define the expected loss function as $\tilde{J}(\boldsymbol{\theta}) = \mathbb{E}_{\boldsymbol{\gamma}}[J(\boldsymbol{\theta}+\boldsymbol{\gamma})]$. We also define $\mathbf{H}(\boldsymbol{\theta})$ as the Hessian matrix for $J(\boldsymbol{\theta})$ and $\tilde{\mathbf{H}}(\boldsymbol{\theta})$ as the Hessian matrix for $\tilde{J}(\boldsymbol{\theta})$. Then the maximum eigenvalue of $\tilde{\mathbf{H}}(\boldsymbol{\theta})$ at any given $\boldsymbol{\theta}$ is less than the maximum eigenvalue of $\mathbf{H}(\boldsymbol{\theta})$ over all $\boldsymbol{\theta} \in \mathbb{R}^n$.*

**Lemma 3.3** *Training a model with Gaussian noise added to parameters is less sensitive to learning rates.*

*Proof.* Let the model parameters be denoted by $\boldsymbol{\theta}$, and the Gaussian noise added to these parameters by $\boldsymbol{\gamma}$. The learning rates of the model with and without Gaussian noise are represented as $\tilde{\eta}$ and $\eta$, respectively, and the corresponding Lipschitz constants are $\tilde{L}$ and $L$. The remaining notations follow from the previous lemmas. From Lemma 3.2, we know that for all $\boldsymbol{\theta} \in \mathbb{R}^N$, the maximum eigenvalue of the Hessian of the smoothed loss function satisfies,

$$\lambda_{\max}(\tilde{H}(\boldsymbol{\theta})) \leq \Lambda_{\max}.$$

Taking the supremum over all $\boldsymbol{\theta}$ and using the properties of the supremum, we obtain

$$\tilde{\Lambda}_{\max} \leq \Lambda_{\max},$$

where $\Lambda_{\max}$ represents the supremum of the maximum eigenvalues of the Hessian of the original loss function $J(\boldsymbol{\theta})$, and $\tilde{\Lambda}_{\max}$ is the supremum of the maximum eigenvalues of the expected Hessian of the smoothed loss function $\tilde{J}(\boldsymbol{\theta})$. From Lemma 3.1, we have the relationship between the Lipschitz constants and the supremum of the Hessian eigenvalues,

$$L = \Lambda_{\max} \quad \text{and} \quad \tilde{L} = \tilde{\Lambda}_{\max}.$$

Therefore, we conclude

$$\tilde{L} \leq L.$$

Since $\tilde{L} \leq L$, and we know that the range of learning rates on which we can guarantee convergence is inversely proportional to the Lipschitz constant, we can say that training with Gaussian noise permits a broader range of allowable learning rates, thus reducing sensitivity to the choice of learning rate during optimization.

### 3.3 Measuring Robustness of a Fine-Tuning Strategy

The previous section discusses the theoretical underpinning of sensitivity of a fine-tuning strategy. Therefore, it is of immense importance to quantify the sensitivity (or robustness) of a fine-tuning method. Robustness in statistical modeling refers to the ability of a model to perform well and provide reliable results across a wide range of conditions and assumptions. In model fine-tuning, a robust strategy can perform reliably under different hyperparameters affecting the fine-tuning process, such as the learning rate of the optimizer, training batch size, etc. Although robustness measures can be defined differently, we resort to statistical measures to quantify the robustness of different fine-tuning strategies. Suppose a model $\mathcal{M}$ is fine-tuned on a given task $\mathcal{T}$ with two different fine-tuning strategies, $\mathcal{S}^{(1)}$ and $\mathcal{S}^{(2)}$, with hyperparameter settings, $\Lambda_1, \Lambda_2, \ldots, \Lambda_k$ and obtains scores $s_1^{(1)}, s_2^{(1)}, \cdots s_k^{(1)}$ and $s_1^{(2)}, s_2^{(2)}, \ldots s_k^{(2)}$, respectively on the validation dataset. Depending on the nature of these scores, we can calculate the metrics to compare the robustness of these fine-tuning strategies. For intrinsic metrics like negative loglikelihood, we calculate *intrinsic robustness* as,

$$\mathcal{R}_{\mathcal{S}^{(1)}} = \frac{1}{\text{med}(\{s_1^{(1)}, s_2^{(1)}, \cdots s_k^{(1)}\})}. \tag{2}$$

Here, *med* is the median of the score distribution. Similarly, for extrinsic metrics like accuracy, we calculate *extrinsic robustness* as,

$$\mathcal{R}_{\mathcal{S}^{(1)}} = \text{med}(\{s_1^{(1)}, s_2^{(1)}, \cdots s_k^{(1)}\}). \tag{3}$$

During model validation, we typically aim for minimizing the intrinsic metrics (for instance, lower validation loss is expected) and maximizing the extrinsic metrics. Therefore, having $\mathcal{R}_{\mathcal{S}^{(1)}} > \mathcal{R}_{\mathcal{S}^{(2)}}$ ensures that the probability of achieving lower intrinsic metric is higher for $\mathcal{S}^{(1)}$ than $\mathcal{S}^{(2)}$, indicating more robustness for $\mathcal{S}^{(1)}$. A similar argument also holds for the extrinsic robustness metric, where a higher score indicates a more robust fine-tuning strategy. As we use the same hyperparameter configurations for comparing different strategies, these robustness metrics allow us to compare different methods without incorporating the variances within the hyperparameters.

| Notation | Description |
|---|---|
| $\boldsymbol{\theta}_i$ | $i$-th column vector in the low-rank matrix |
| $\boldsymbol{\mu}_i$ | $i$-th column mean vector |
| $\mathcal{W}$ | Wishart Distirbution |
| $V$ | Scale matrix of Wishart distribution |
| $\boldsymbol{\Sigma}$ | Sampled covariance matrix |
| Dir | Dirichlet Distirbution |
| $\boldsymbol{\alpha}$ | Dirichlet distribution concentration parameter |
| $\boldsymbol{\Pi}$ | Mixture weights vector |
| $\pi_k$ | $k$-th mixture weight |
| N | Number of inner Monte Carlo samples |
| M | Number of outer Monte Carlo samples |
| $\boldsymbol{W}_{stochastic}$ | Stochastic component sampled from the Monte Carlo process |
| $\epsilon$ | Sample scaler |

Table 2: Glossary of all the mathematical notations used with `MonteCLoRA`.

### 3.4 Bayesian Inference and Monte Carlo Estimation Methods

In the Bayesian treatment of machine learning and deep learning models, we are often concerned with the predictive distribution and aim to incorporate prior information into our estimates. Bayesian methods typically regularize the weight space of the model, resulting in a robust reparameterization of the models. With Bayesian formulation, we can parameterize the probability distribution of a model parameter $\boldsymbol{\theta}$ as,

$$P(\boldsymbol{\theta}) = \int P(\boldsymbol{\theta} \mid \beta)P(\beta)d\beta \tag{4}$$

where $\beta$ denotes the parameters that we learn to represent the distribution of the weight space of the model. The integral is often highly intractable due to the complex and high-dimensional nature of the distributions involved. Monte Carlo estimation provides a simple computational algorithm for calculating computationally intractable statistics to calculate analytically.

## 4 Proposed Methodology

In this section, we introduce our proposed method, `MonteCLoRA`, a **Monte C**arlo enhanced **Lo**w-**R**ank **A**daptation for fine-tuning LLMs. It learns a posterior distribution for each low-rank $\boldsymbol{A}$ parameter and estimates a robust and unbiased posterior estimator. With Monte Carlo estimation, `MonteCLoRA` prevents mode collapse of the low-rank parameters, instead learns to sample from a basin of optima for improved generalization and robustness. Figure 3 illustrates our overall methodology. Table 2 describes all the mathematical notations to describe `MonteCLoRA`.

### 4.1 Gaussian Factorization of Weight Space

Learning a posterior distribution over the entire weight space of a model is a complex and computationally demanding task. To address this, we adopt a factorization assumption, where we presume that the model's weight space distribution can be decomposed into the product of the distributions across all layers. This approach decouples the distribution learning of one weight matrix from others, enabling efficient parameter estimation.

Our primary focus is on low-rank matrices, though this methodology extends to full-rank matrices. Suppose we have a low-rank weight matrix $\boldsymbol{\theta} \in \mathbb{R}^{r \times n_{\mathrm{in}}}$. We denote the probability distribution of its weight space as $P(\boldsymbol{\theta})$. A key assumption is that each of the $n_{\mathrm{in}}$ column vectors ($\boldsymbol{\theta}_i \in \mathbb{R}^r$) independently follows a Gaussian distribution with a shared covariance matrix but with distinct means,

$$P(\boldsymbol{\theta}) = P(\boldsymbol{\theta}_1, \boldsymbol{\theta}_2, \ldots, \boldsymbol{\theta}_{n_{\mathrm{in}}}) = \prod_{i=1}^{n_{\mathrm{in}}} P(\boldsymbol{\theta}_i). \tag{5}$$

We parameterize the distribution of each $\boldsymbol{\theta}_i$ by assuming that, given a covariance matrix $\boldsymbol{\Sigma} \in \mathbb{R}^{r \times r}$, each $\boldsymbol{\theta}_i$ is generated from $\mathcal{N}(\boldsymbol{\mu}_i, \boldsymbol{\Sigma})$, with $\boldsymbol{\mu}_i$ being the corresponding column mean vector. Using a common covariance

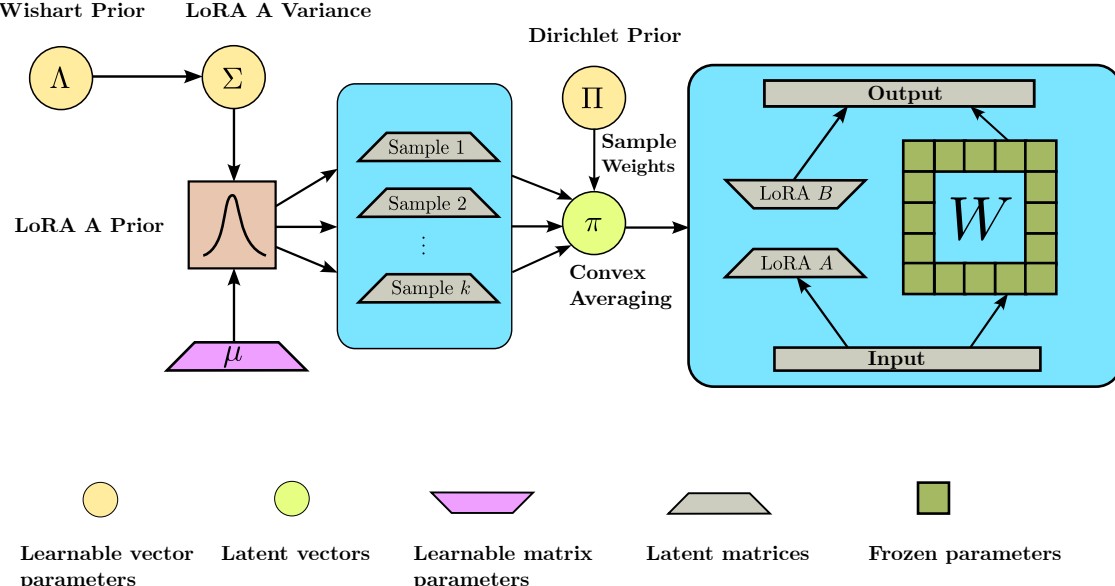

Figure 3: Overview of `MonteCLoRA` for Bayesian Estimation of LoRA Parameters. `MonteCLoRA` models the LoRA-A matrix as a mixture of Gaussians centered at a shared learnable mean $\mu$ with covariance matrices $\Sigma$ sampled from a Wishart prior. Mixture weights $\pi$ are drawn from a Dirichlet prior. **(1)** A diagonal scale matrix $V$ defines the Wishart prior from which covariances $\Sigma$ are sampled. **(2)** $k$ samples are drawn from $\mathcal{N}(\mu, \Sigma)$ to produce candidate LoRA-A matrices. **(3)** A convex combination of these samples is computed using Dirichlet weights $\pi \sim \text{Dir}(\alpha)$. This approach introduces only $\mathcal{O}(r + N)$ additional parameters per LoRA layer.

matrix allows for efficient posterior learning without adding a significant number of additional parameters. The covariance matrix is not a fixed parameter; instead, it is sampled from a Wishart distribution (for details on the distribution, check Appendix C) with learnable priors. The learnable prior parameters enable us to explore the low-rank parametric weight space, leading to more robust parameter estimates for the model. We define $P(\boldsymbol{\theta}_i)$ as a weighted sum of $N$ samples drawn from a common multivariate Gaussian distribution:

$$P(\boldsymbol{\theta}_i) = \sum_{k=1}^{N} \pi_k \cdot \boldsymbol{\theta}_{i,k}, \quad \text{s.t.} \quad \sum_{k=1}^{N} \pi_k = 1, \tag{6}$$

where each sample $\boldsymbol{\theta}_{i,k} \sim \mathcal{N}(\boldsymbol{\mu}_i, \boldsymbol{\Sigma})$ is independently drawn from a single Gaussian distribution with a shared mean $\boldsymbol{\mu}_i$ and covariance $\boldsymbol{\Sigma}$. The mixture weights $\boldsymbol{\Pi} = (\pi_1, \pi_2, \ldots, \pi_N)$ are drawn from a Dirichlet distribution, $\text{Dir}(\boldsymbol{\alpha})$, with concentration parameter $\boldsymbol{\alpha} = (\alpha_1, \alpha_2, \ldots, \alpha_N)$. Unlike a traditional Gaussian mixture model, this formulation does not involve separate components with different means or covariances; instead, it constructs a convex combination of multiple samples from the same Gaussian prior.

The mean vector $\boldsymbol{\mu}_i$ is initialized randomly at the start of training (essentially the LoRA $\boldsymbol{A}$ matrices) and is subsequently learned. The covariance matrix $\boldsymbol{\Sigma}$ is sampled from a Wishart prior distribution $\mathcal{W}_r(\boldsymbol{V}, n_{\text{in}})$, where $\boldsymbol{V}$ is a learnable diagonal scale matrix, $n_{\text{in}}$ is the degrees of freedom, and $r$ is the LoRA rank. Given the independence assumption across dimensions, we restrict $\boldsymbol{V}$ to be diagonal. Therefore, using Equations 5 and 6, we can write the joint density as,

$$P(\boldsymbol{\theta}, \boldsymbol{\Pi}, \boldsymbol{\Sigma}) = \prod_{i=1}^{n_{\text{in}}} \left( \sum_{k=1}^{N} \pi_k \cdot \mathcal{N}(\boldsymbol{\theta}_k; \boldsymbol{\mu}_i, \boldsymbol{\Sigma}) \right) \cdot \text{Dir}(\boldsymbol{\Pi}; \boldsymbol{\alpha}) \cdot \mathcal{W}_r(\boldsymbol{\Sigma}; \boldsymbol{V}, n_{\text{in}}). \tag{7}$$

Note that different LoRA parameters may use distinct covariance matrices learned with different Wishart scale matrices.

## 4.2 Monte Carlo Estimation of LoRA Parameters

Given that $\mathbf{\Pi}$ and $\mathbf{\Sigma}$ are independent, we can simplify the Monte Carlo estimation for the expectation of $\boldsymbol{\theta}_i$. The expectation of $\boldsymbol{\theta}_i$ is,

$$\mathbb{E}[\boldsymbol{\theta}_i] = \int \boldsymbol{\theta}_i \, P(\boldsymbol{\theta}_i|\boldsymbol{\mu_i}, \mathbf{\Pi}, \mathbf{\Sigma}) \, P(\mathbf{\Pi}) \, P(\mathbf{\Sigma}) \, d\boldsymbol{\theta}_i \, d\mathbf{\Pi} \, d\mathbf{\Sigma}. \tag{8}$$

In the Monte Carlo formulation, we approximate this integral by sampling from the distributions of $\mathbf{\Pi}$ and $\mathbf{\Sigma}$ and then computing the expectation over $\boldsymbol{\theta}_i$, given each sampled pair $(\mathbf{\Pi}, \mathbf{\Sigma})$ and a fixed $\boldsymbol{\mu_i}$ .

1. **Sample $\mathbf{\Pi}$ and $\mathbf{\Sigma}$**

   (a) Draw $M$ samples $\mathbf{\Pi}^{(j)} \sim \text{Dir}(\boldsymbol{\alpha})$, where $j = 1, \dots, M$.
   (b) Draw $M$ samples $\mathbf{\Sigma}^{(j)} \sim \mathcal{W}_r(\boldsymbol{V}, n_{\text{in}})$, where $j = 1, \dots, M$.

2. **Sample $\boldsymbol{\theta}_i$ given $\mathbf{\Pi}$, $\mathbf{\Sigma}$ and $\boldsymbol{\mu_i}$**

   For each pair $(\mathbf{\Pi}^{(j)}, \mathbf{\Sigma}^{(j)})$, sample $N$ values of $\boldsymbol{\theta}_i^{(k,j)} \sim P(\boldsymbol{\theta}_i|\boldsymbol{\mu_i}, \mathbf{\Pi}^{(j)}, \mathbf{\Sigma}^{(j)})$, where $k = 1, \dots, N$.

3. **Compute the Monte Carlo Estimate**

   Using the samples, the expectation $\mathbb{E}[\boldsymbol{\theta}_i]$ is estimated as:

$$\boldsymbol{E}[\boldsymbol{\theta}_i] \approx \frac{1}{M} \sum_{j=1}^{M} \frac{1}{N} \sum_{k=1}^{N} \boldsymbol{\theta}_i^{(k,j)}. \tag{9}$$

The inner sum $\frac{1}{N} \sum_{k=1}^{N} \boldsymbol{\theta}_i^{(k,j)}$ approximates the conditional expectation, $\mathbb{E}[\boldsymbol{\theta}_i|\boldsymbol{\mu_i}, \mathbf{\Pi}^{(j)}, \mathbf{\Sigma}^{(j)}]$. The outer sum averages over the prior samples of $\mathbf{\Pi}$ and $\mathbf{\Sigma}$ and a fixed $\boldsymbol{\mu_i}$, estimating the marginal expectation $\mathbb{E}[\boldsymbol{\theta}_i]$. This two-layer Monte Carlo approach approximates the integral by leveraging the independence of $\mathbf{\Pi}$ and $\mathbf{\Sigma}$ and sampling $\boldsymbol{\theta}_i$ conditioned on each sampled pair $(\mathbf{\Pi}, \mathbf{\Sigma})$. The independence assumption permits separate sampling for each $\boldsymbol{\theta}_i$, combined into a final sampled weight matrix.

---

**Algorithm 1** `MonteCLoRA` Estimation of LoRA Parameters

---

**Require:** $n_{\text{in}}$: Input feature size
**Require:** $n_{\text{out}}$: Output feature size
**Require:** $N$: Number of mixture components
**Require:** $\boldsymbol{V} \in \mathbb{R}^{n_{\text{out}} \times n_{\text{out}}}$: Wishart Distribution Prior (Trainable Diagonal Matrix)
**Require:** $\boldsymbol{\alpha} \in \mathbb{R}^N$: Dirichlet Distribution Prior (Trainable Vector)
**Require:** $\boldsymbol{\mu} \in \mathbb{R}^{n_{\text{in}} \times n_{\text{out}}}$: Weight matrix of the linear layer (Trainable Dense Matrix)
**Require:** $\boldsymbol{\epsilon}$: Sample Scaler
$\quad \mathbf{\Sigma} \sim \mathcal{W}_{n_{\text{out}}}(\boldsymbol{V}, n_{\text{in}})$ $\qquad\qquad\qquad\qquad\qquad$ ▷ Sample from Wishart Distribution
$\quad \pi_1, \pi_2, \dots, \pi_N \sim \text{Dir}(\boldsymbol{\alpha})$ $\qquad\qquad\qquad\qquad\quad$ ▷ Sample from Dirichlet Distribution
$\quad \boldsymbol{S}^{(1)}, \boldsymbol{S}^{(2)}, \cdots, \boldsymbol{S}^{(N)} \sim \mathcal{N}(\boldsymbol{0}, \mathbf{\Sigma})$ $\qquad\quad$ ▷ Sampling from multivariate Gaussian distribution
$\quad \boldsymbol{W}_{\texttt{MonteCLoRA}} \leftarrow \boldsymbol{\mu} + \boldsymbol{\epsilon} * \sum_{i=1}^{N} \pi_k \cdot \boldsymbol{S}^{(k)}$ $\qquad\quad$ ▷ Compute the `MonteCLoRA` estimate
$\quad \boldsymbol{\mu} \leftarrow \boldsymbol{W}_{\texttt{MonteCLoRA}}$ $\qquad\qquad$ ▷ Update the weight matrix with the `MonteCLoRA` estimate

---

Algorithm 1 formalizes the forward method of `MonteCLoRA` for each LoRA parameter initialized with[3] $\boldsymbol{\mu}$ and $M = 1$. We scale the sample variance before adding to $\boldsymbol{\mu}$ by the sample scaler ($\epsilon$) to have a finer control over changes in the model weights. The learnable parameters in the above approach are the LoRA parameters, Wishart distribution scale matrix $\boldsymbol{V}$ and the Dirichlet concentration parameter $\boldsymbol{\alpha}$. The Wishart distribution scale is a diagonal matrix of order $n_{\text{out}}$, and the Dirichlet concentration is another vector of size $N$. Hence, for each `MonteCLoRA` layer, we add $n_{\text{out}} + N$ parameters only. When this is applied to LoRA

---

[3] Please note that we use $\boldsymbol{\mu}$ and $\boldsymbol{\theta}$ to refer to the LoRA weights but in different contexts.. In contexts of distributions we use $\boldsymbol{\mu}$ and in contexts of learnable parameter we use $\boldsymbol{\theta}$.

down projection matrix, the number of extra parameters is just $r + N$, and we introduce only a fraction of parameters $\mathcal{O}(\frac{r+N}{n_{in} \times r})$ obtained from LoRA, which is approximately $\mathcal{O}(1)$ as $r \ll min(n_{in}, n_{out})$, and $N$ are usually of same order which in turn is a only small fraction of total parameters for a layer. Hence, the total number of parameters over the LoRA parameters is very small and $\mathcal{O}(1)$ compared to LoRA parameters.

## 4.3 Why do we use a mixture of Gaussian

Using a single Gaussian for all of the low-rank parameters forces the entire posterior to be unimodal and centered around a single mean-covariance estimate. In contrast, drawing multiple samples from the same Gaussian and then forming a convex combination, allows `MonteCLoRA` to explore different modes in the weight space and avoid collapsing onto a single point estimate.

Moreover, by drawing multiple samples, `MonteCLoRA` obtains a Monte Carlo approximation that reduces variance, providing a robust estimate of the original LoRA parameters, under a Bayesian framework. As proved in Lemma 3.3, adding noise to the weights during training makes a neural model less sensitive to different learning rates during training. Therefore, using the Gaussian parameterization, `MonteCLoRA` provides smoother and more reliable convergence during training the low-rank parameters.

## 4.4 Theoretical Results

This section presents the theoretical results involving `MonteCLoRA`, for demonstrating the effectiveness of Bayesian parameterization for robust fine-tuning of LLMs. We show that `MonteCLoRA` derives an unbiased and robust estimate of the true model parameters and how it influences and stabilizes the final output obtained by the fine-tuned model.

**Lemma 4.1** *Sample scale factor $\epsilon$ shifts the final output proportionally.*

*Proof.* Consider a simplified version of the model where we ignore activations. The neural network can be represented as a product of multiple weight matrices, $\boldsymbol{\theta}_1, \boldsymbol{\theta}_2, \ldots, \boldsymbol{\theta}_n$. The output for an input $\boldsymbol{X}$ is given by,

$$\boldsymbol{Y} = \boldsymbol{\theta}_1 \boldsymbol{\theta}_2 \cdots \boldsymbol{\theta}_n \boldsymbol{X}.$$

Now, we consider the parameterization used by `MonteCLoRA`. Each weight matrix $\boldsymbol{\theta}_i$ includes a scaled random variable added to it. We represent random variables as $\hat{\boldsymbol{\phi}}_i$, and the scaling factor is $\epsilon$. The output for an input $\boldsymbol{X}$ in this case becomes,

$$\bar{\boldsymbol{Y}} = (\boldsymbol{\theta}_1 + \epsilon \hat{\boldsymbol{\phi}}_1)(\boldsymbol{\theta}_2 + \epsilon \hat{\boldsymbol{\phi}}_2) \cdots (\boldsymbol{\theta}_n + \epsilon \hat{\boldsymbol{\phi}}_n) \boldsymbol{X}.$$

Expanding the terms and neglecting $\mathcal{O}(\epsilon^2)$ terms (this assumption is valid as we typically use $\epsilon \in \mathcal{O}(10^{-3})$), we obtain

$$\bar{\boldsymbol{Y}} = \boldsymbol{\theta}_1 \boldsymbol{\theta}_2 \cdots \boldsymbol{\theta}_n \boldsymbol{X} + \epsilon \left( \hat{\boldsymbol{\phi}}_1 \boldsymbol{\theta}_2 \cdots \boldsymbol{\theta}_n + \boldsymbol{\theta}_1 \hat{\boldsymbol{\phi}}_2 \boldsymbol{\theta}_3 \cdots \boldsymbol{\theta}_n + \cdots + \boldsymbol{\theta}_1 \boldsymbol{\theta}_2 \cdots \boldsymbol{\theta}_{n-1} \hat{\boldsymbol{\phi}}_n \right) \boldsymbol{X} + \mathcal{O}(\epsilon^2)$$

$$= \boldsymbol{Y} + \epsilon \sum_{i=1}^{n} \left( \boldsymbol{\theta}_1 \cdots \boldsymbol{\theta}_{i-1} \hat{\boldsymbol{\phi}}_i \boldsymbol{\theta}_{i+1} \cdots \boldsymbol{\theta}_n \right) \boldsymbol{X}.$$

Taking the norm, we obtain,

$$\|\bar{\boldsymbol{Y}} - \boldsymbol{Y}\| = \epsilon \left\| \sum_{i=1}^{n} \left( \boldsymbol{\theta}_1 \cdots \boldsymbol{\theta}_{i-1} \hat{\boldsymbol{\phi}}_i \boldsymbol{\theta}_{i+1} \cdots \boldsymbol{\theta}_n \right) \right\| \|\boldsymbol{X}\|.$$

Assuming $\|\boldsymbol{X}\| > 0$

$$\frac{\|\bar{\boldsymbol{Y}} - \boldsymbol{Y}\|}{\|\boldsymbol{X}\|} = \epsilon \left\| \sum_{i=1}^{n} \left( \boldsymbol{\theta}_1 \cdots \boldsymbol{\theta}_{i-1} \hat{\boldsymbol{\phi}}_i \boldsymbol{\theta}_{i+1} \cdots \boldsymbol{\theta}_n \right) \right\|.$$

This expression shows that for an input $\boldsymbol{X}$, the output shift is directly proportional to the sample scaling factor $\epsilon$.

**Lemma 4.2** *The final model output estimated by* `MonteCLoRA` *is an unbiased estimator of the original model output.*

*Proof.* An important conclusion from the parameterization shown in the previous section is the expected value of $\bar{\boldsymbol{Y}}$. Since each weight matrix is assumed to be independent, we have,

$$\mathbb{E}[\bar{\boldsymbol{Y}}] = \mathbb{E}\left[(\boldsymbol{\theta}_1 + \epsilon\hat{\boldsymbol{\phi}}_1)\right]\mathbb{E}\left[(\boldsymbol{\theta}_2 + \epsilon\hat{\boldsymbol{\phi}}_2)\right]\cdots\mathbb{E}\left[(\boldsymbol{\theta}_n + \epsilon\hat{\boldsymbol{\phi}}_n)\right]\boldsymbol{X}$$

$$\mathbb{E}[\bar{\boldsymbol{Y}}] = (\boldsymbol{\theta}_1 + \epsilon\mathbb{E}[\hat{\boldsymbol{\phi}}_1])(\boldsymbol{\theta}_2 + \epsilon\mathbb{E}[\hat{\boldsymbol{\phi}}_2])\cdots(\boldsymbol{\theta}_n + \epsilon\mathbb{E}[\hat{\boldsymbol{\phi}}_n])\boldsymbol{X}$$

Since each $\hat{\boldsymbol{\phi}}_i$ is a weighted sum of samples from a Gaussian distribution with zero mean, we have $\mathbb{E}[\hat{\boldsymbol{\phi}}_i] = 0$. Therefore,

$$\mathbb{E}[\bar{\boldsymbol{Y}}] = \boldsymbol{\theta}_1\boldsymbol{\theta}_2\cdots\boldsymbol{\theta}_n\boldsymbol{X} = \boldsymbol{Y}.$$

Hence, in expectation, the model's prediction is not dependent on the sample scaler. This property is beneficial during inference. We run the model at inference time like the standard LoRA model, utilizing the fine-tuned weights guided by `MonteCLoRA` enhancements. As a result, inference times remain the same because we do not need to perform sampling during inference.

**Lemma 4.3** *The estimator described in Equation 9 is an unbiased estimator for the expected posterior defined in Equation 8.*

*Proof.* Let $\hat{\boldsymbol{\theta}}_i$ denote the estimator in Equation 9. To show that the Monte Carlo estimator is an unbiased estimator of the true expectation, we compute $\mathbb{E}[\hat{\boldsymbol{\theta}}_i]$ and demonstrate that it equals $\mathbb{E}[\boldsymbol{\theta}_i]$. For fixed $\boldsymbol{\Pi}^{(j)}$ and $\boldsymbol{\Sigma}^{(j)}$, the expectation of $\frac{1}{N}\sum_{k=1}^{N}\boldsymbol{\theta}_i^{(k,j)}$ over the samples $\boldsymbol{\theta}_i^{(k,j)}$ is,

$$\mathbb{E}\left[\frac{1}{N}\sum_{k=1}^{N}\boldsymbol{\theta}_i^{(k,j)} \mid \boldsymbol{\Pi}^{(j)}, \boldsymbol{\Sigma}^{(j)}\right] = \mathbb{E}[\boldsymbol{\theta}_i \mid \boldsymbol{\Pi}^{(j)}, \boldsymbol{\Sigma}^{(j)}].$$

By the law of large numbers, as $N \to \infty$, the sample mean $\frac{1}{N}\sum_{i=k}^{N}\boldsymbol{\theta}_i^{(k,j)}$ converges to the conditional expectation $\mathbb{E}[\boldsymbol{\theta}_i \mid \boldsymbol{\Pi}^{(j)}, \boldsymbol{\Sigma}^{(j)}]$. Next, we consider the expectation of the outer sum over the samples $\boldsymbol{\Pi}^{(j)}$ and $\boldsymbol{\Sigma}^{(j)}$ as,

$$\mathbb{E}\left[\frac{1}{M}\sum_{j=1}^{M}\mathbb{E}[\boldsymbol{\theta}_i \mid \boldsymbol{\Pi}^{(j)}, \boldsymbol{\Sigma}^{(j)}]\right] = \mathbb{E}_{\boldsymbol{\Pi},\boldsymbol{\Sigma}}[\mathbb{E}[\boldsymbol{\theta}_i \mid \boldsymbol{\Pi}, \boldsymbol{\Sigma}]].$$

Using the law of total expectation, we have

$$\mathbb{E}_{\boldsymbol{\Pi},\boldsymbol{\Sigma}}[\mathbb{E}[\boldsymbol{\theta}_i \mid \boldsymbol{\Pi}, \boldsymbol{\Sigma}]] = \mathbb{E}[\boldsymbol{\theta}_i].$$

Therefore, as $M \to \infty$, the outer sum $\frac{1}{M}\sum_{j=1}^{M}\mathbb{E}[\boldsymbol{\theta}_i \mid \boldsymbol{\Pi}^{(j)}, \boldsymbol{\Sigma}^{(j)}]$ converges to $\mathbb{E}[\boldsymbol{\theta}_i]$, which is the original integral. Hence, in expectation, the Monte Carlo estimator $\hat{\boldsymbol{\theta}}_i$ converges to the true integral $\mathbb{E}[\hat{\boldsymbol{\theta}}_i] = \mathbb{E}[\boldsymbol{\theta}_i]$. This demonstrates that, given sufficiently large $N$ and $M$, the Monte Carlo estimate will approximate the desired expectation $\mathbb{E}[\boldsymbol{\theta}_i]$.

**Lemma 4.4** *The estimator described in Equation 9 is a robust estimator for the integral in Equation 8.*

*Proof.* To find the covariance of the Monte Carlo estimator described in Equation 9, we need to analyze both the inner and outer layers of sampling over $\theta_i$, $\boldsymbol{\Pi}$, and $\boldsymbol{\Sigma}$. The variance of this estimator will depend on the variability from both layers. We first decompose the covariance using the law of total covariance; we can decompose the variance of $\hat{\boldsymbol{\theta}}_i$ as follows,

$$\text{Cov}(\hat{\boldsymbol{\theta}}_i) = \mathbb{E}_{\boldsymbol{\Pi},\boldsymbol{\Sigma}}\left[\text{Cov}_{\boldsymbol{\theta}_i}\left(\frac{1}{N}\sum_{k=1}^{N}\theta_i^{(k,j)} \mid \boldsymbol{\Pi}, \boldsymbol{\Sigma}\right)\right] + \text{Cov}_{\boldsymbol{\Pi},\boldsymbol{\Sigma}}\left(\mathbb{E}_{\boldsymbol{\theta}_i}\left[\frac{1}{N}\sum_{k=1}^{N}\boldsymbol{\theta}_i^{(k,j)} \mid \boldsymbol{\Pi}, \boldsymbol{\Sigma}\right]\right).$$

- The first term, $\mathbb{E}_{\boldsymbol{\Pi},\boldsymbol{\Sigma}}\left[\mathrm{Cov}_{\boldsymbol{\theta}_i}\left(\frac{1}{N}\sum_{i=1}^{N}\boldsymbol{\theta}^{(i,j)}\mid\boldsymbol{\Pi},\boldsymbol{\Sigma}\right)\right]$, represents the expected conditional covariance given $\boldsymbol{\Pi}$ and $\boldsymbol{\Sigma}$.

- The second term, $\mathrm{Cov}_{\boldsymbol{\Pi},\boldsymbol{\Sigma}}\left(\mathbb{E}_{\boldsymbol{\theta}_i}\left[\frac{1}{N}\sum_{i=1}^{N}\boldsymbol{\theta}^{(i,j)}\mid\boldsymbol{\Pi},\boldsymbol{\Sigma}\right]\right)$, represents the covariance due to the variability in $\boldsymbol{\Pi}$ and $\boldsymbol{\Sigma}$.

For fixed $\boldsymbol{\Pi}$ and $\boldsymbol{\Sigma}$, the covariance of the average $\frac{1}{N}\sum_{k=1}^{N}\boldsymbol{\theta}_i^{(k,j)}$ is,

$$\mathrm{Cov}_{\boldsymbol{\theta}_i}\left(\frac{1}{N}\sum_{k=1}^{N}\boldsymbol{\theta}_i^{(k,j)}\mid\boldsymbol{\Pi},\boldsymbol{\Sigma}\right)=\frac{1}{N}\mathrm{Cov}_{\boldsymbol{\theta}_i}(\boldsymbol{\theta}_i\mid\boldsymbol{\Pi},\boldsymbol{\Sigma}),$$

where $\mathrm{Cov}_{\boldsymbol{\theta}_i}(\boldsymbol{\theta}_i\mid\boldsymbol{\Pi},\boldsymbol{\Sigma})$ is the conditional covariance of $\boldsymbol{\theta}_i$ given $\boldsymbol{\Pi}$ and $\boldsymbol{\Sigma}$. So, the first term becomes,

$$\mathbb{E}_{\boldsymbol{\Pi},\boldsymbol{\Sigma}}\left[\frac{1}{N}\mathrm{Cov}(\boldsymbol{\theta}_i\mid\boldsymbol{\Pi},\boldsymbol{\Sigma})\right]=\frac{1}{N}\mathbb{E}_{\boldsymbol{\Pi},\boldsymbol{\Sigma}}[\mathrm{Cov}_{\boldsymbol{\theta}_i}(\boldsymbol{\theta}_i\mid\boldsymbol{\Pi},\boldsymbol{\Sigma})].$$

The conditional covariance can be computed as,

$$\mathrm{Cov}(\boldsymbol{\theta}\mid\boldsymbol{\Pi},\boldsymbol{\Sigma})=\sum_i\pi_i^2\boldsymbol{\Sigma}.$$

On taking expectation over $\boldsymbol{\Pi}$ and $\boldsymbol{\Sigma}$, we obtain,

$$\frac{1}{N}\mathbb{E}_{\boldsymbol{\Pi},\boldsymbol{\Sigma}}\left[\sum_i\pi_i^2\boldsymbol{\Sigma}\right]=\frac{1}{N}\sum_i\mathbb{E}_{\boldsymbol{\Pi}}[\pi_i^2]\boldsymbol{\Sigma}=\frac{\boldsymbol{\Sigma}}{N}\sum_i\mathbb{E}_{\boldsymbol{\Pi}}[\pi_i^2].$$

The second term in the covariance decomposition accounts for the variability of the conditional expectation $\mathbb{E}[\boldsymbol{\theta}\mid\boldsymbol{\Pi},\boldsymbol{\Sigma}]$ due to the sampling of $\boldsymbol{\Pi}$ and $\boldsymbol{\Sigma}$. We can write this term as,

$$\mathrm{Cov}_{\boldsymbol{\Pi},\boldsymbol{\Sigma}}\left(\mathbb{E}\left[\boldsymbol{\theta}\mid\boldsymbol{\Pi},\boldsymbol{\Sigma}\right]\right)=\frac{1}{M}\mathrm{Cov}_{\boldsymbol{\Pi},\boldsymbol{\Sigma}}\left(\mathbb{E}[\boldsymbol{\theta}\mid\boldsymbol{\Pi},\boldsymbol{\Sigma}]\right).$$

The expectation of $\boldsymbol{\theta}$ given $\boldsymbol{\Pi}$ and $\boldsymbol{\Sigma}$ is just the mean of the distribution, which is independent of $\boldsymbol{\Pi}$ and $\boldsymbol{\Sigma}$. Hence the second term goes to zero. Putting it all together, the covariance of the estimator $\hat{\boldsymbol{\theta}}_i$ is,

$$\mathrm{Cov}(\hat{\boldsymbol{\theta}}_i)=\frac{\boldsymbol{\Sigma}}{N}\mathbb{E}_{\boldsymbol{\Pi}}[\sum_i\pi_i^2].$$

Therefore, as $N\to\infty$, the covariance reduces, making the estimator robust.

## 4.5 Training Objectives

### 4.5.1 Reparameterization Losses

We incorporate a series of Kullback-Leibler divergence (KLD) losses into the objective function to regularize the learned prior distributions. These losses are crucial for shaping the model's latent weight space and preventing overfitting. For each sampling process within the model, a corresponding KLD loss is added. The sum of all these KLD losses across layers is scaled by a KL divergence weight parameter $\eta$ and incorporated into the final loss.[4]

- **Multivariate Gaussian KL Divergence Loss.** In `MonteCLoRA`, the distribution to be optimized is a multivariate Gaussian distribution with parameters $\left(\boldsymbol{\mu},\hat{\boldsymbol{\Sigma}}\right)$ and it is optimized against a multivariate Gaussian with parameters $(\boldsymbol{\mu},\boldsymbol{I})$. We refer to the distribution to be optimized as $P$ and the standard normal as $Q$. The KL divergence loss can be computed as,

$$\mathrm{KL}_{\mathcal{N}}=\frac{1}{2}\left(\mathrm{tr}(\hat{\boldsymbol{\Sigma}})-\ln|\hat{\boldsymbol{\Sigma}}|-n_{\mathrm{out}}\right). \tag{10}$$

---

[4]Derivation of the KL divergence losses are furnished in the Appendix D.

- **Wishart KL Divergence Loss.** We optimise the Wishart Distribution of `MonteCLoRA`, $\mathcal{W}_{n_{\text{out}}}(\mathbf{V}, n_{\text{in}})$, against the standard Wishart distribution, $\mathcal{W}_{n_{\text{out}}}(\boldsymbol{I}, n_{\text{in}})$. We keep the degree of freedom and dimensionality the same to focus on optimizing the scale matrix. We refer to the distribution to be optimized as $P$ and the standard distribution as $Q$. The KL divergence can be calculated in closed form as,

$$\text{KL}_{\mathcal{W}} = \frac{1}{2}\left(n_{\text{in}}\left(-\ln|\mathbf{V}|\right) + n_{\text{in}}, \text{tr}(\mathbf{V}) - n_{\text{in}}n_{\text{out}}\right). \tag{11}$$

- **Dirichlet KL Divergence Loss.** We have $N$-Dimensional Dirichlet random vectors following the distributions ($P$ and $Q$) with parameters $\boldsymbol{\alpha}_1$ and $\boldsymbol{\alpha}_2$, respectively, where $\boldsymbol{\alpha}_1, \boldsymbol{\alpha}_2 \in \mathbb{R}^N$. In `MonteCLoRA`, we take $\boldsymbol{\alpha}_2$ as a constant vector with all the same values. The KL divergence of $P$ from $Q$ is given by,

$$\text{KL}_{\mathcal{D}} = \ln\frac{\Gamma\left(\sum_{i=1}^{N}\alpha_{1i}\right)}{\Gamma\left(\sum_{i=1}^{N}\alpha_{2i}\right)} + \sum_{i=1}^{N}\ln\frac{\Gamma(\alpha_{2i})}{\Gamma(\alpha_{1i})} + \sum_{i=1}^{N}(\alpha_{1i} - \alpha_{2i})\left[\psi(\alpha_{1i}) - \psi\left(\sum_{j=1}^{N}\alpha_{1j}\right)\right]. \tag{12}$$

Here, $\Gamma$ is the gamma, and $\psi$ is the digamma (logarithmic derivative of gamma) function.

### 4.5.2 Cooperative Loss

To ensure maximal participation from each of the $N$ mixture component, we compute a *cooperative loss*. For mixture weights $\pi_1, \pi_2, \ldots, \pi_N$ obtained from the Dirichlet samples, the cooperative loss is calculated as $\sum_{i=1}^{N}\pi_i^2$. To promote higher cooperation, we minimize the cooperative loss,

$$\text{L}_{\mathcal{C}} = \sum_{i=1}^{N}\pi_i{}^2. \tag{13}$$

**Lemma 4.5** *Cooperative loss defined in Equation 13 is minimized when $\pi_i = \frac{1}{N} \; \forall i \in [1, N]$.*

*Proof.* Since the samples from a dirichlet distribution sum to one, we have $\sum_{i=1}^{N}\pi_i = 1$, we can rewrite $\text{L}_{\mathcal{C}}$ as,

$$\pi_1^2 + \pi_2^2 + \cdots + (1 - \sum_{i=1}^{N-1}\pi_i)^2 = 1 + 2\sum_{i=1}^{N-1}\pi_i^2 - 2\sum_{i=1}^{N-1}\pi_i + 2\sum_{i=1}^{N-1}\sum_{j\neq i}\pi_i\pi_j.$$

Therefore, for $\frac{\partial \text{L}_{\mathcal{C}}}{\partial \pi_i} = 0$, we get,

$$4\pi_i - 2 + 2\sum_{j\neq i}\pi_j = 0$$

$$\implies 2\pi_i = 2 - 2\sum_{i=1}^{N-1}\pi_i$$

$$\implies 2\pi_i = 2\pi_N.$$

Hence, $\pi_1 = \pi_2 = \cdots = \pi_N = \frac{1}{N}$. As $\frac{\partial^2 \text{L}_{\mathcal{C}}}{\partial^2 \pi_i} = 4 > 0$, $\frac{1}{N}$ is the minima.

**Lemma 4.6** *Minimizing cooperative loss defined in Equation 13 is equivalent to minimizing entropy of mixture of Gaussian defined in Equation 6.*

*Proof.* Using the fact that $\boldsymbol{\theta}_{i,k} \sim \mathcal{N}(\boldsymbol{\mu}_i, \boldsymbol{\Sigma})$ and entropy of $\mathcal{N}(\boldsymbol{\mu}_i, \boldsymbol{\Sigma}) = c + \frac{1}{2}\log det(\Sigma)$, for some suitable constant $c$, we get

$$entropy(P(\boldsymbol{\theta}_i)) = entropy(\sum_{k=1}^{N} \pi_k \cdot \mathcal{N}(\boldsymbol{\mu}_i, \boldsymbol{\Sigma})) \equiv \sum_{k=1}^{N} \frac{1}{2} \log \pi_k^2 \cdot det(\Sigma) = \frac{N}{2} \log det(\Sigma) + \frac{1}{2} \sum_{k=1}^{N} \log \pi_k^2. \quad (14)$$

Monotonicity of $\log(x)$ suggests that minimizing $\sum_{k=1}^{N} \pi_k^2$ minimizes the entropy of $P(\boldsymbol{\theta}_i)$.

Therefore, minimizing the cooperative loss $\mathrm{L}_{\mathcal{C}}$ reduces the uncertainty of the mixture of Gaussian learned in Equation 6, leading to robust estimate of the LoRA parameters.

### 4.5.3 Final Loss Function

Combining all the training objectives, the final loss function can be defined as,

$$\mathcal{L} = \sum_{(x,y) \in Z} \sum_{t=1}^{|y|} \log, P_{\theta_0 + \Delta\theta(\Theta,\zeta)} (y_t \mid x, y_{<t}) + \eta \cdot \frac{(\mathrm{KL}_{\mathcal{D}} + \mathrm{KL}_{\mathcal{W}} + \mathrm{KL}_{\mathcal{N}})}{N_{\texttt{MonteCLoRA}}} + \mathrm{L}_{\mathcal{C}}.$$

Here, $N_{\texttt{MonteCLoRA}}$ denotes the total number of $\texttt{MonteCLoRA}$ layers. The division by $N_{\texttt{MonteCLoRA}}$ and the KLD loss weight $\eta$ serve to normalize and scale down the contribution of the KL divergence losses, ensuring balanced training dynamics.

## 5 Experiments

### 5.1 Datasets and Tasks

To evaluate the effectiveness of $\texttt{MonteCLoRA}$, we conduct thorough empirical study across two different ranges of tasks – natural language understanding (NLU) and natural language generation (NLG).

For NLU, we use five tasks from the GLUE (Wang et al., 2018) and SuperGLUE (Wang et al., 2019) benchmarks, namely MRPC (Dolan & Brockett, 2005), CoLA (Warstadt et al., 2019), RTE (Dagan et al., 2005; Haim et al., 2006; Giampiccolo et al., 2007; Bentivogli et al., 2009), WiC (Pilehvar & Camacho-Collados, 2018) and BoolQ (Clark et al., 2019). Details of these datasets can be found in Appendix B. These gold standard datasets for these tasks contain separate train and dev (also called validation) split, where we use the train dataset to fine-tune LLMs and dev dataset for evaluation. For these tasks, we consider the intrinsic evaluation metric – negative loglikelihood (NLL) and extrinsic evaluation metric – accuracy.

On commonsense NLG, we consider six commonsense reasoning tasks – PiQA (Bisk et al., 2020), Social (Sap et al., 2019), WinoGrande (Sakaguchi et al., 2021), ARC-easy, ARC-challenge (Clark et al., 2018) and OpenBookQA (Mihaylov et al., 2018). Details of these tasks are mentioned in Appendix B. For these tasks, we use the Commonsense15K dataset (Hu et al., 2023). This dataset is particularly curated for instruction fine-tuning of LLMs and comprises of subsets of training samples from different commonsense reasoning tasks. The final output for these NLG tasks contains the answer to a multiple-choice question and is evaluated using accuracy. We conduct further evaluation on instruction following tasks on mathematical reasoning and code generation. For evaluation on mathematical reasoning tasks, we consider the GSM8k (Cobbe et al., 2021) dataset. For code generation, we construct a dataset from the publicly available Magicoder-OSS-Instruct-75K corpus (Wei et al., 2024). Specifically, we filter for Python examples whose tokenized sequence length is below 512 tokens, and randomly sample 10,000 such data points to form a consistent and lightweight training set. To assess model performance, we evaluate on the official test split of GSM8k for math reasoning and HumanEval benchmark (Chen et al., 2021), a widely-used dataset for functional code synthesis performance. Details of these tasks are mentioned in Appendix B.

### 5.2 Models

For NLU, we use a pre-trained RoBERTA-base (Liu et al., 2019) (110M parameters) model. For commonsense generative tasks, we use the pre-trained LLaMA-1-7B (Touvron et al., 2023) model, and for math and coding

| Type | Hyperparameter | RoBERTa-base | LLaMA-1-7B |
|---|---|---|---|
| Static | LoRA $r$ | 8 | 32 |
| | LoRA $\alpha$ | 16 | 64 |
| | Max Sequence Length | 256 | 256 |
| | Learning Rate Scheduler | Linear | Linear |
| | Epochs | 20 | 3 |
| Tunable | Batch Size | $\{8, 32, 64\}$ | $\{8, 16, 32\}$ |
| | Learning Rate | $\{3 \times 10^{-4}, 5 \times 10^{-5}, 1 \times 10^{-5}\}$ | $\{3 \times 10^{-4}, 3 \times 10^{-5}\}$ |

Table 3: Static and tunable hyperparameters used in fine-tuning RoBERTa-base and LLaMA-1-7B models.

| Hyperparameter | Value |
|---|---|
| Sample Scaler ($\epsilon$) | $5 \times 10^{-3}$ , $2.5 \times 10^{-5}$(LLaMA-3.2) |
| KL Loss Weight ($\eta$) | $1 \times 10^{-5}$ |
| Dirichlet Prior ($\boldsymbol{\alpha}$) | 1 |
| Mixture Components ($N$) | 4 |

Table 4: Hyperparameters used for `MonteCLoRA`.

tasks we use Llama-3.2-3B-Instruct (Grattafiori et al., 2024a). All the pre-trained model weights are obtained from Huggingface (Wolf et al., 2020).

## 5.3 Baselines

Apart from full fine-tuning and LoRA (Hu et al., 2022) fine-tuning strategies, we compare the performance of `MonteCLoRA` against the competitive parameter-efficient fine-tuning methods.

- **AdaLoRA.** Zhang et al. (2023) improved upon vanilla LoRA by dynamically adjusting the ranks of the rank-decomposition matrices, facilitating the allocation of more capacity to important weights and reducing it for less significant ones.

- **DoRA.** Liu et al. (2024b) improved on LoRA by decomposing the pre-trained weight into two components, magnitude and direction, for fine-tuning. By employing LoRA for directional updates to efficiently minimize the number of trainable parameters, DoRA makes sure to enhance the learning capacity and training stability of LoRA while avoiding any additional inference overhead.

- **Laplace LoRA.** Yang et al. (2024) brought a Bayesian approach to LoRA by applying a Laplace approximation to the posterior over LoRA parameters to overcome uncertainty and overconfidence and ensure better model calibration.

- **LoRA+.** Hayou et al. (2024) extended the original LoRA by introducing additional trainable components such as bias and layer norm parameters, along with a gating mechanism to selectively control LoRA updates. This enables improved flexibility and performance, especially in tasks requiring nuanced adaptation, while maintaining a low parameter footprint.

- **Prompt Tuning.** Lester et al. (2021) proposed a lightweight fine-tuning strategy where a small set of continuous trainable vectors (soft prompts) are prepended to the input embeddings of a frozen pre-trained model. This method drastically reduces the number of trainable parameters while achieving strong task performance, especially in language understanding tasks.

- **IA$^3$.** Liu et al. (2022a) introduced IA$^3$ (Input-Output-Attention Adaptation) as a parameter-efficient fine-tuning method by inserting learnable scaling vectors into the key, value, and intermediate projection layers of transformer blocks. By modulating the flow of information through attention and MLP layers, IA$^3$ enables effective adaptation with minimal parameter updates.

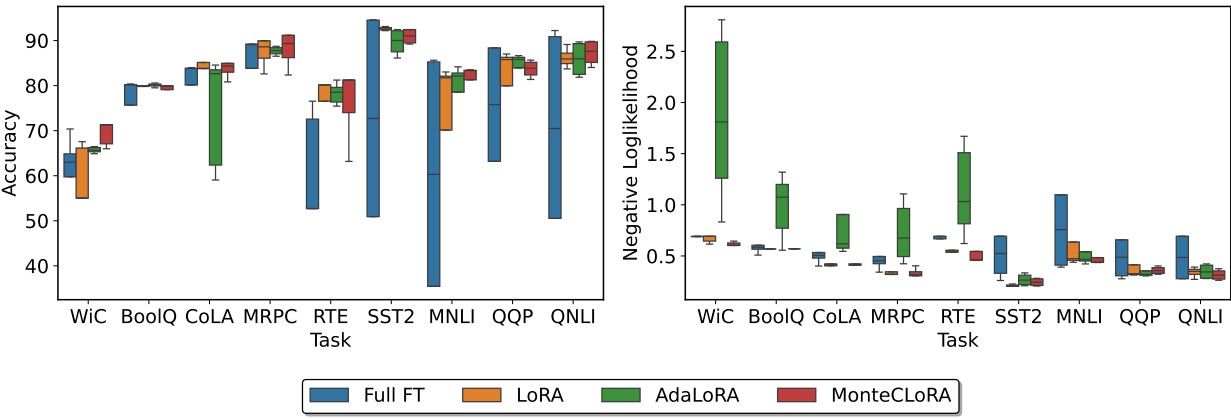

Figure 4: Distribution of accuracies and negative loglikelihood of RoBERTa-base on different GLUE tasks with different fine-tuning strategies.

## 5.4 Hyperparameters

As described in Section 1, LLM fine-tuning strategies are very sensitive to hyperparameters. For the reproducibility of our study, we describe the hyperparameters used in `MonteCLoRA` and the baseline on different tasks. Table 3 contains the static and tunable hyperparameters used for all the fine-tuning methods with RoBERTa and LLaMA models. Static hyperparameters are used for all the model-specific training tasks and are the same for all the fine-tuning strategies. We tune the optimizer learning rate and the training batch size for the robustness studies for different strategies. Table 4 reports the hyperparameters specific to `MonteCLoRA`. The hyperparameters are the default ones used in our method and are used in all the NLU and NLG tasks. In the subsequent sections, we discuss the importance of these hyperparameters in greater detail. All our experiments were conducted on NVIDIA A100-80GB GPUs that had access to the CUDA 12.5 environment.

## 6 Experimental Results

In this section, we discuss the results obtained from the empirical study with the NLU and NLG tasks.

### 6.1 Evaluation on NLU Tasks

Figure 4 highlights the distribution of accuracy (extrinsic metric) and negative loglikelihood (intrinsic metric) calculated on the validation dataset of the GLUE and SuperGLUE tasks with the RoBERTa-base model. We compare `MonteCLoRA` with full fine-tuning (denoted as Full FT), LoRA and AdaLoRA. The distribution spread (calculated as best - worse score) is observed to be highest for full fine-tuning, with the average spread being 0.30 and 28.7, respectively, on the intrinsic and extrinsic metrics, significantly higher than all the PEFT methods. Additionally, this spread in accuracy is highly pronounced in the larger GLUE tasks, highlighting the sensitivity of full fine-tuning on hyperparameters particularly for larger and semantically more challenging tasks. Among the low-rank adaptation methods, AdaLoRA has the highest average spread of 0.61 with the intrinsic metric, whereas it is 0.10 for `MonteCLoRA`, which highlights the ability of `MonteCLoRA` to calibrate output confidence under different hyperparameter configurations. LoRA has NLL spread 0.18, which is less than that of AdaLoRA, but still considerably higher than that of `MonteCLoRA`. `MonteCLoRA` also fares much better than the other methods in terms of average accuracy spreads. It is the least for `MonteCLoRA`, with a value of 5.40 and worst for LoRA with a value of 19.44. On smaller tasks like WiC, LoRA demonstrates a 7 points higher spread than `MonteCLoRA`. `MonteCLoRA` shows higher stability in terms of both metrics, justifying its effectiveness in maintaining stability in performance and confidence in its predictions.

| Method | Robustness | MRPC | CoLA | RTE | WiC | BoolQ | SST2 | QNLI | QQP | MNLI | Average |
|---|---|---|---|---|---|---|---|---|---|---|---|
| Full FT | | 2.21 | 1.98 | 1.45 | 1.45 | 1.71 | 1.91 | 2.06 | 2.05 | 1.32 | 1.79 |
| LoRA | Intrinsic ↑ | 3.11 | 2.38 | 1.86 | 1.49 | **1.76** | **4.81** | 2.87 | 3.06 | 2.11 | **2.61** |
| AdaLoRA | | 1.48 | 1.61 | 0.97 | 0.55 | 0.93 | 3.79 | 2.91 | **3.12** | 2.14 | 1.94 |
| MonteCLoRA | | __3.16__ | __2.43__ | __2.18__ | __1.63__ | 1.75 | 4.09 | __3.20__ | 2.81 | __2.27__ | __2.61__ |
| Full FT | | 89.1 | 83.8 | 52.7 | 63.0 | 80.1 | 72.7 | 84.0 | 81.4 | 60.3 | 74.1 |
| LoRA | Extrinsic ↑ | 88.6 | 83.8 | 80.1 | 60.6 | 80.0 | **92.6** | 85.3 | 85.5 | 81.7 | 82.0 |
| AdaLoRA | | 87.7 | 82.6 | 78.5 | 65.7 | **80.2** | 90.0 | 86.0 | **85.8** | 82.1 | 82.1 |
| MonteCLoRA | | __89.3__ | __84.3__ | __81.2__ | __69.2__ | 79.9 | 91.0 | __89.6__ | 85.0 | __83.3__ | __83.6__ |

Table 5: Robustness of different fine-tuning strategies with RoBERTa-base on GLUE and SuperGLUE tasks. We underline the tasks where `MonteCLoRA` achieves higher robustness score than LoRA fine-tuning.

| Method | Metric | MRPC | CoLA | RTE | WiC | BoolQ | SST2 | QNLI | QQP | MNLI | Average |
|---|---|---|---|---|---|---|---|---|---|---|---|
| Full FT | | 89.2 | 84.0 | 76.5 | 70.4 | 80.4 | **94.6** | **92.2** | **88.4** | **85.6** | **84.6** |
| LoRA | Accuracy ↑ | 89.9 | **85.1** | 80.1 | 67.5 | 80.0 | 93.1 | 89.1 | 87.0 | 83.0 | 83.9 |
| AdaLoRA | | 88.7 | 84.5 | **81.2** | 67.7 | **80.6** | 92.4 | 89.7 | 86.6 | 84.2 | 84.0 |
| MonteCLoRA | | **91.2** | 84.9 | **81.2** | **71.3** | 79.9 | 92.4 | 89.8 | 85.7 | 83.5 | 84.4 |
| Full FT | | 0.34 | **0.40** | 0.60 | 0.68 | 0.51 | 0.26 | 0.28 | **0.28** | **0.39** | 0.42 |
| LoRA | NLL ↓ | 0.32 | **0.40** | 0.54 | 0.62 | **0.49** | **0.20** | 0.27 | 0.31 | 0.44 | 0.40 |
| AdaLoRA | | 0.42 | 0.54 | 0.62 | 0.83 | 0.55 | 0.21 | 0.28 | 0.30 | 0.42 | 0.46 |
| MonteCLoRA | | **0.31** | 0.41 | **0.46** | **0.60** | 0.50 | 0.21 | **0.26** | 0.32 | 0.44 | **0.39** |

Table 6: Comparison of different fine-tuning strategies with RoBERTa-base on GLUE and SuperGLUE tasks. We report the highest accuracy and lowest negative loglikelihood (NLL) obtained across different hyperparameter configurations. The best strategy is highlighted in **bold** and the second best strategy is underlined for each task.

Table 5 shows the intrinsic and extrinsic robustness defined in Section 3.3 for different fine-tuning methods. The above discussion confirms that `MonteCLoRA` tends to have lesser chances of having abysmally bad intrinsic or extrinsic scores (outliers), unlike FFT or LoRA. The robustness metrics highlight the method's stability in terms of maintaining the probability of having good performance. We observe the highest average intrinsic robustness value of 2.61 both with `MonteCLoRA` and LoRA. AdaLoRA demonstrates the lowest intrinsic robustness among the low-rank methods, 26% lower than `MonteCLoRA`. `MonteCLoRA` also exhibits the highest extrinsic robustness of 83.6, which is 1.6 points higher than LoRA, and 1.5 points higher than AdaLoRA. We perform Wilcoxon signed-rank test to statistically validate whether `MonteCLoRA` is more robust than the other baselines. The rank tests conclude that `MonteCLoRA` is more robust than LoRA and its variants, in terms of extrinsic ($p$-value 0.08) robustness metrics. Full fine-tuning displays lowest robustness values, which is more pronounced for the bigger tasks. It has the lowest extrinsic robustness of 74.1, and intrinsic robustness of 1.79 indicating its inability to generalize well across different NLU tasks. The underperformance of full fine-tuning also suggests that training the full language model on all downstream tasks is inefficient and could be ineffective for certain tasks. Particularly on tasks like RTE, full fine-tuning can diverge and exhibit very poor performance on validation. Remarkably, `MonteCLoRA` achieves higher intrinsic and extrinsic robustness than LoRA and its variant in six out of nine tasks, emphasizing its ability to showcase robustness across different tasks and training configurations.

Table 6 reports the highest accuracy and the lowest negative loglikelihood (correspondingly, highest loglikelihood) achieved by different fine-tuning methods on NLU tasks. Interestingly, the variance among the baselines is significantly lower when the best results are concerned. On average, the best accuracy obtained with full fine-tuning is 84.6%, which is higher than LoRA and AdaLoRA. `MonteCLoRA` comes a close second with 84.4% accuracy, which the best among the low-rank methods, demonstrating its superiority in achieving better performance across different hyperparameter configurations. It also achieves the lowest NLL among all the baselines, indicating its superiority in generalization across multiple tasks.

We compare `MonteCLoRA` against the contemporary Bayesian post-hoc methods on the NLU tasks in Table 7. The Bayesian methods such as maximum a-posteriori (MAP), Monte Carlo dropout (MC Dropout), and Laplace-LoRA (LA) offer flexible solutions to applying Bayesian treatment to LoRA fine-tuning in a post-

| Metric | Method | MRPC | CoLA | RTE | WiC | BoolQ | Average |
|---|---|---|---|---|---|---|---|
| Accuracy ↑ | MonteCLoRA (best) | **91.2** | **84.9** | **81.2** | **71.3** | **79.9** | **81.7** |
| | MonteCLoRA (median) | **89.3** | **84.3** | **81.2** | **69.2** | **79.9** | **80.8** |
| | MAP | 86.4 | 81.8 | 70.9 | 63.9 | 77.2 | 76.0 |
| | MC Dropout (Gal & Ghahramani, 2016) | 87.1 | 82.6 | 72.4 | 68.8 | 76.6 | 77.5 |
| | Checkpoint Ensemble (Chen et al., 2017) | 86.3 | 81.4 | 71.8 | 64.7 | 77.2 | 76.3 |
| | Temp (Guo et al., 2017) | 86.5 | 81.8 | 72.6 | 65.4 | 77.3 | 76.7 |
| | LLLA (Yang et al., 2024) | 86.4 | 81.8 | 72.6 | 65.3 | 77.4 | 76.7 |
| | LA (Yang et al., 2024) | 86.4 | 81.7 | 72.6 | 65.4 | 77.4 | 76.7 |
| NLL ↓ | MonteCLoRA (best) | **0.31** | 0.41 | **0.46** | **0.60** | 0.50 | **0.46** |
| | MonteCLoRA (median) | **0.32** | 0.41 | **0.46** | **0.61** | 0.57 | **0.47** |
| | MAP | 0.66 | 0.50 | 0.76 | 1.00 | 0.54 | 0.69 |
| | MC Dropout (Gal & Ghahramani, 2016) | 0.39 | 0.39 | 0.58 | 0.72 | 0.50 | 0.52 |
| | Checkpoint Ensemble (Chen et al., 2017) | 0.44 | 0.49 | 0.57 | 0.63 | 0.53 | 0.53 |
| | Temp (Guo et al., 2017) | 0.32 | 0.40 | 0.54 | 0.62 | 0.49 | 0.47 |
| | LLLA (Yang et al., 2024) | 0.33 | 0.44 | 0.56 | 0.78 | 0.51 | 0.52 |
| | LA (Yang et al., 2024) | 0.34 | 0.39 | 0.54 | 0.62 | 0.48 | 0.47 |

Table 7: Comparison of different Bayesian post-hoc methods applied to pre-trained RoBERTa-base model and LoRA fine-tuning on various GLUE and SuperGLUE tasks. Results of baselines apart from MonteCLoRA are obtained from Yang et al. (2024). The bold results indicate the cases where MonteCLoRA performs better than the other Bayesian baselines.

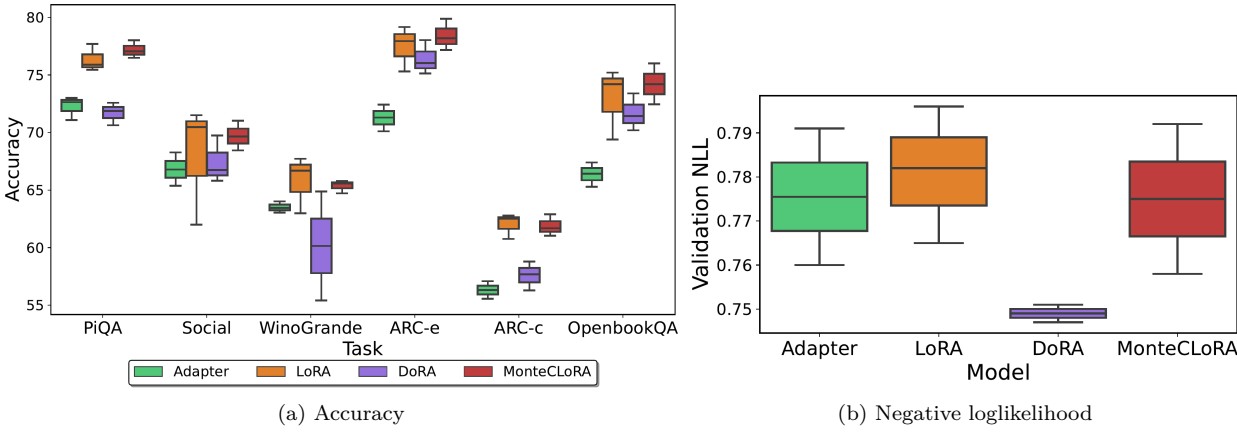

(a) Accuracy               (b) Negative loglikelihood

Figure 5: Distribution of test accuracies and validation negative loglikelihood of LLaMA-1-7B on different commonsense reasoning tasks with different fine-tuning strategies.

| Method | PiQA (↑) | Social(↑) | WinoGrande(↑) | ARC-e(↑) | ARC-c(↑) | OpenbookQA(↑) | Average(↑) |
|---|---|---|---|---|---|---|---|
| Adapter | 73.0 | 68.3 | 64.0 | 72.4 | 57.1 | 67.4 | 67.0 |
| LoRA | 77.7 | **71.5** | **67.7** | 79.2 | 62.8 | 75.2 | **72.3** |
| DoRA | 72.6 | 69.7 | 64.9 | 78.0 | 58.8 | 73.4 | 69.6 |
| MonteCLoRA | **78.0** | 71.0 | 65.8 | **79.9** | **62.9** | **76.0** | **72.3** |

Table 8: Performance of different fine-tuning strategies with pre-trained LLaMA-1-7B on generative tasks.

hoc manner (i.e., can be used after the LoRA method is trained). Our comparative analysis suggests that MonteCLoRA can perform better than the existing Bayesian post-hoc methods with fewer training steps. As argued by Yang et al. (2024), these methods require longer training steps for calibrating output probabilities generated in the fine-tuning phase. Even when we consider the median performance of MonteCLoRA, it still achieves 3.2% higher accuracy than the best baseline, Monte Carlo Dropout. In terms of the negative loglikelihood, MonteCLoRA achieves the best performance among all these Bayesian post-hoc methodologies.

## 6.2 Evaluation on Commonsense NLG Tasks

We follow a similar empirical study for the generative tasks as in the NLU tasks. Figure 5 highlights the distribution of test accuracy and the validation NLL scores. Contrarily to the NLU tasks, the generative

| Method | PiQA(↑) | Social(↑) | WinoGrande(↑) | ARC-e(↑) | ARC-c(↑) | OpenbookQA(↑) | Average(↑) |
|---|---|---|---|---|---|---|---|
| Adapter | 72.6 | 66.8 | 63.4 | 71.3 | 56.3 | 66.4 | 66.1 |
| LoRA | 75.9 | **70.5** | **66.7** | 77.9 | **62.5** | 74.2 | **71.3** |
| DoRA | 71.9 | 66.7 | 60.1 | 76.0 | 57.7 | 71.4 | 67.3 |
| MonteCLoRA | **77.0** | 69.6 | 65.6 | **78.2** | 61.7 | **74.3** | 71.1 |

Table 9: Extrinsic robustness of different fine-tuning strategies with LLaMA-1-7B on zero-shot generative tasks.

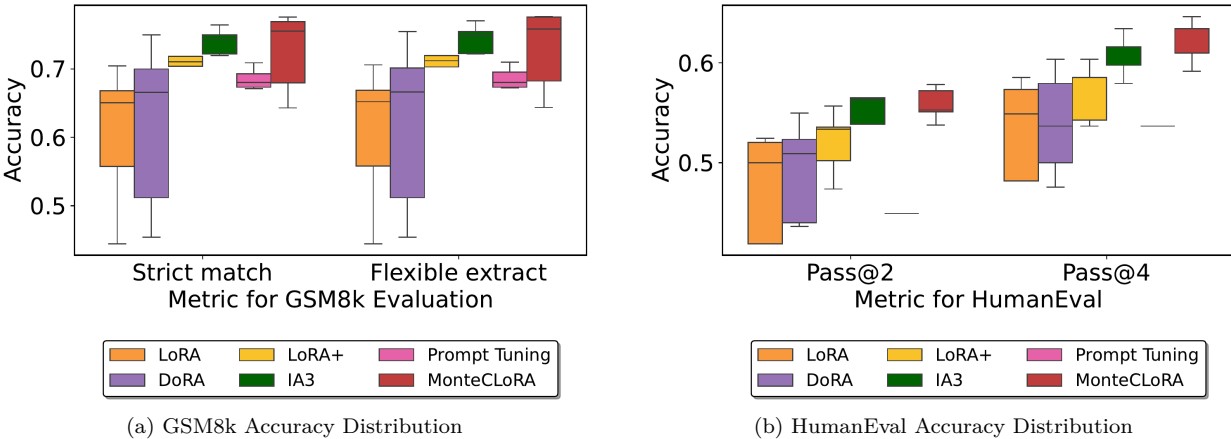

(a) GSM8k Accuracy Distribution      (b) HumanEval Accuracy Distribution

Figure 6: Distribution of accuracies obtained by LLaMA-3.2-3B in GSM8k and HumanEval tasks for different PEFT strategies.

experiments are more intriguing as the extrinsic metrics are calculated on a test dataset in a zero-shot manner, i.e., the model might not be aware of the test data distribution during the fine-tuning phase. In terms of the intrinsic metric (NLL), DoRA has the tiniest spread of 0.004 points, whereas both LoRA and MonteCLoRA have a modest spread of 0.03. However, with extrinsic metric, the spread with MonteCLoRA is only 2.2, significantly lower than LoRA (4.6) and DoRA (4.0). Remarkably, on tasks like WinoGrande, the spread with DoRA and LoRA can be as large as 9.5; however, with MonteCLoRA, the spread remains meagre for all the tasks, ensuring higher stability.

In Table 8, we report best accuracies achieved by the different fine-tuning methods on generative tasks. We observe that MonteCLoRA and LoRA exhibit a similar performance on the zero-shot generative tasks. The average accuracies obtained with MonteCLoRA and LoRA remains 72.3, higher than Adapter and DoRA. Out of six tasks, on three tasks, MonteCLoRA achieves better accuracy than LoRA. Out of the remaining three tasks, only in WinoGrande, the margin between LoRA and MonteCLoRA is significant. LoRA and MonteCLoRA are marginally similar for the remaining two tasks. In terms of the robustness metrics (c.f. Table 9), LoRA performs slightly better than MonteCLoRA. Particularly on ARC-challenge and WinoGrande, LoRA achieves ∼ 1 point better robustness than MonteCLoRA. However, it is worth noting that the long-tail nature of the accuracy distribution of LoRA makes these results less reliable and less robust. On the other hand, MonteCLoRA balances good performance with excellent stability, overmining its effectiveness in robust fine-tuning.

## 6.3 Evaluation of NLG Tasks with LLaMA-3.2-3B-Instruct on GSM8k and HumanEval

We further comprehensively evaluate various PEFT methods on GSM8k and HumanEval datasets using LLaMA-3.2-3B-Instruct as the base model. These tasks are strong benchmarks for mathematical reasoning and code synthesis, respectively. Both the evaluations were done using LM-evaluation-harness (Gao et al., 2024).

| PEFT | SM (max)↑ | FE (max)↑ | SM (ext. robust.)↑ | FE (ext. robust.)↑ |
|---|---|---|---|---|
| LoRA | 0.70 | 0.71 | 0.65 | 0.65 |
| DoRA | 0.75 | 0.75 | 0.67 | 0.67 |
| LoRA+ | 0.76 | 0.76 | 0.71 | 0.71 |
| IA$^3$ | 0.76 | 0.77 | 0.74 | 0.75 |
| Prompt Tuning | 0.71 | 0.71 | 0.68 | 0.68 |
| MonteCLoRA | **0.78** | **0.78** | **0.75** | **0.76** |

Table 10: Comparison of PEFT methods on GSM8k using SM (strict match) and FE (flexible match) metrics with robustness evaluation.

| PEFT | pass@2 (max)↑ | pass@4 (max)↑ | pass@2 (ext. robust.)↑ | pass@4 (ext. robust.)↑ |
|---|---|---|---|---|
| LoRA | 0.52 | 0.58 | 0.50 | 0.55 |
| DoRA | 0.55 | 0.60 | 0.51 | 0.54 |
| LoRA+ | 0.56 | 0.60 | 0.53 | 0.58 |
| IA$^3$ | 0.56 | 0.63 | **0.56** | **0.62** |
| Prompt Tuning | 0.45 | 0.54 | 0.45 | 0.54 |
| MonteCLoRA | **0.58** | **0.65** | 0.55 | 0.61 |

Table 11: Comparison of PEFT methods on HumanEval using pass@2 and pass@4 metrics with robustness evaluation.

On GSM8k, we adopt the exact match (EM) metric to evaluate the strict correctness of the model's generated answers. Due to the presence of varied generation formats, we apply two filtering strategies: *strict-match*, which enforces precise output formatting, and *flexible-extract*, which tolerates verbose reasoning and isolates the final answer via regex-based parsing. This dual evaluation scheme enables a nuanced understanding of the model's reasoning accuracy and its robustness to formatting constraints. Table 10 shows that MonteCLoRA consistently outperforms the baselines across all metrics. It achieves the highest strict match score (0.78) and flexible extract score (0.78) while attaining top robustness scores, demonstrating its superior generalization to clean and noisy test environments. IA$^3$ and LoRA+ also show strong performance, though with slightly lower robustness margins. Prompt Tuning trails behind the adaptation-based methods in both maximum and robust scores. The spread analysis (Figure 6a) indicates that MonteCLoRA has a significantly tighter distribution than LoRA and DoRA, whereas it is behind IA3 and prompt. On strict match metric, LoRA shows a performance spread of up to 2.6 points, 2.4 for DoRA, and MonteCLoRA's spread remains within 1.26 points, which is 51% less compared to LoRA and 47.5% less compared to DoRA, underscoring its robustness and reliability.

On the HumanEval benchmark (Table 11), which measures functional correctness in code generation using pass@k metrics, MonteCLoRA again emerges as the most performant method. With maximum pass@2 and pass@4 scores of 0.58 and 0.65, respectively, it outperforms all other PEFT strategies. Its robustness (median) pass@2 and pass@4 scores remain high (0.55 and 0.61) and beat all LoRA family baselines, indicating strong consistency in generating functionally correct code across different learning rates. IA$^3$ and LoRA+ perform competitively, and IA$^3$ beats MonteCLoRA by a small margin in robustness. Prompt Tuning, in contrast, underperforms significantly, suggesting its limited capacity for structured generative tasks like code synthesis. As seen in Figure 6b, the robustness spread in HumanEval is similarly minimized with MonteCLoRA. While LoRA and DoRA suffer from a spread of 3.8 and 1.1 points, MonteCLoRA maintains a tight band of 0.4 around its peak, reducing spread by 89% for LoRA and 63% for DoRA, respectively, further reinforcing its stability.

In conclusion, while traditional methods such as LoRA, LoRA+, and IA$^3$ provide competitive baselines, MonteCLoRA balances top-tier accuracy and robustness, solidifying its effectiveness for fine-tuning complex NLG tasks involving symbolic reasoning and code generation. Its consistently narrow spread in both max and robust metrics makes it a promising candidate for real-world applications requiring stability under distributional shifts.

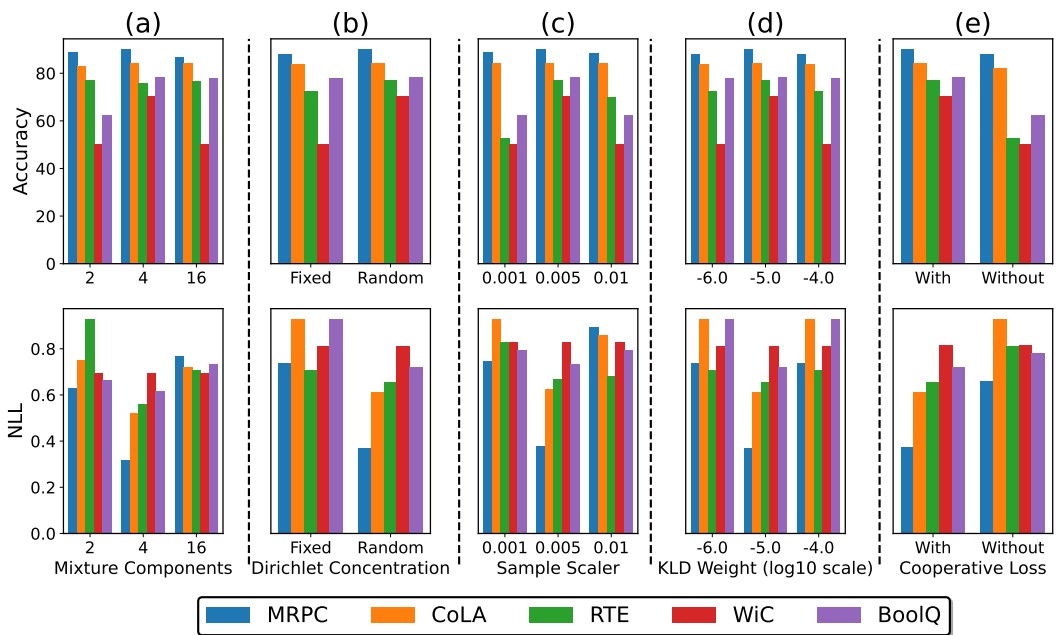

Figure 7: Ablation of `MonteCLoRA` with RoBERTa-base on different GLUE tasks with respect to (a) the number of mixture components $N$, (b) Dirichlet concentration parameter $\boldsymbol{\alpha}$, (c) Sample scaler $\epsilon$, (d) KLD weight ($\eta$), and (e) cooperative loss.

## 6.4  Ablation Study

We study the importance of different components of `MonteCLoRA` with thorough ablation analysis. We use the NLU tasks with the RoBERTa-base coupled with `MonteCLoRA` fine-tuning for this study. Figure 7 highlights the performance of `MonteCLoRA` by changing the hyperparameters, including the number of mixture components ($N$), Dirichlet concentration parameter ($\boldsymbol{\alpha}$), sample scaler ($\epsilon$), KLD loss weight ($\eta$), and the importance of cooperative loss defined in Equation 13.

**Effect of the Number of Mixture Components.** Lemma 4.4 suggests that `MonteCLoRA` estimates a robust estimator with a large number of mixture components. However, in Figure 7(a), we observe that `MonteCLoRA` can achieve robust performance even with only four mixture components. Increasing the mixtures to 16 has very minimal impact on the accuracy and the loglikelihood of the fine-tuned model. Having too less number of mixture components could lead to underfitting, hurting the performance of the fine-tuned model.

**Importance of Dirichlet Concentration Initialization.** To understand the impact of initialization of the Dirichlet concentration, we perform an ablation where instead of initializing the value with $\boldsymbol{\alpha} = 1$, we use a random vector $\mathcal{U}(0,1)^N$, where $\mathcal{U}$ is a uniform distribution. We call this ablation 'random' Dirichlet concentration. The results are particularly surprising (c.f. Figure 7(b)), where we observe an average $> 2\%$ improvement in accuracy with random initialization. Random prior can improve the accuracy on tasks like RTE by a margin of 4%. Random initialization of Dirichlet concentration parameters allows the model for more exploration, learning the mixture dynamics better.

**Sample Scaler and KLD Loss Weight for Balancing Exploration-Exploitation.** In Algorithm 1, we described the importance of the sample scaler $\epsilon$ to dynamically control the exploration of different optimization basins of the low-rank parameters. The ablation results in Figure 7(c) highlights an interesting pattern: with a moderate $\epsilon = 0.005$, `MonteCLoRA` achieves the best validation accuracy. The results emphasize the importance of balancing exploration and exploitation for a stable convergence of the fine-tuned model. In RTE, WiC, and BoolQ, a high $\epsilon$ can diverge the model, whereas a low $\epsilon$ can lead to overfitting the training

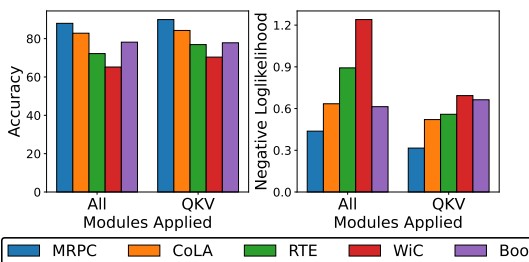 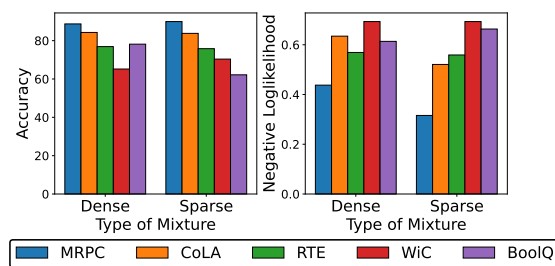

Figure 8: Performance of `MonteCLoRA` at different modules of RoBERTa-base.

Figure 9: importance of a mixture of samples in `MonteCLoRA`.

data, both impacting the validation performance. A similar observation is found with the KLD loss weight ($\eta$) in Figure 7(d). A high KLD loss weight indicates more regularization towards the prior distributions, whereas a low KLD loss weight indicates assigning more importance to the likelihood term, relaxing the prior conditions. Having a moderate $\eta = 10^{-5}$ leads to better and more stable performance. In fact, $\eta = 10^{-6}$ or $\eta = 10^{-4}$ leads to almost same performance drop of $1\% - 4\%$, for different tasks.

**Cooperative Loss for Better Allocation of Mixture Importance.** Another critical component of `MonteCLoRA` training is the cooperative loss. Without cooperative loss, the mixture importance values remain unconstrained and independent. A sense of cooperation between different mixture components ensures that the dynamics of the complex low-rank parameterization is captured. Moreover, with cooperative loss, the importance of different mixture components can be allocated proportionately, maintaining the system's entropy. In Figure 7(e), we see a drastic $> 3\%$ performance drop without the cooperative loss component, indicating its importance in achieving good generalization.

**Flexibility of `MonteCLoRA` on Different LLM Components.** Figure 8 shows the difference in performance on GLUE tasks when `MonteCLoRA` is applied on all the parameters (including attention query, attention key, attention value, attention output, intermediate output and layer output modules) versus when applied only on attention parameters (query, key and value parameters). The analysis strikes an interesting pattern where we see improvement in performance for larger tasks and marginal performance drop for smaller tasks when `MonteCLoRA` is applied to all parameters. In larger tasks like BoolQ, applying `MonteCLoRA` on all parameters improves the negative loglikelihood and accuracy by 0.5 and 0.3 points, respectively. However, on smaller tasks, enabling `MonteCLoRA` to only attention parameters improves accuracy by a margin of 2% on average. This phenomenon can be justified using the fact that Transformer MLP (which includes the intermediate and layer output modules) blocks encourage sparse activation (Li et al., 2022); therefore, having dense mixtures can adversely affect the fine-tuned model. Moreover, sparse connections are more important on smaller tasks, where overfitting could be a key issue. Therefore, the dense mixture introduced by `MonteCLoRA` is not particularly effective with MLP blocks, specifically when applied to smaller downstream tasks.

**Introducing Sparsity in `MonteCLoRA`.** The previous analysis highlights that a sparser mixture of Gaussian components could be deemed important while fine-tuning LLMs on smaller tasks. To encourage sparsity in the mixture components, we perform an ablation experiment where we choose only one component. Formally, given a mixture weights $\{\pi_1, \pi_2, \cdots \pi_N\}$, we only activate the component $k$, where $k = \arg\max_i\{\pi_i\}$. Therefore, the updated mixture weight values become,

$$\pi_i' = \begin{cases} 1, & \text{if } i = k \\ 0, & \text{otherwise} \end{cases}$$

Figure 9 highlights the results with dense (the default mixture of components) and sparse components on NLU tasks. We observe a significant accuracy improvement on two smaller tasks, MRPC and WiC (1.2% and 5%, respectively) with the sparse mixture. The results illustrate the flexibility of adapting `MonteCLoRA`

on different complexities of tasks, which most existing low-rank parameterization-based fine-tuning methods fail to exhibit.

**Post-hoc Abilities of `MonteCLoRA`.**   In Section 2, we described the post-hoc Bayesian methods for robust and calibrated reparameterization of fine-tuned LLMs. Therefore, to assess the effectiveness of `MonteCLoRA` in post-hoc execution, we perform experiments where we first fine-tune the LLM with only the $\boldsymbol{\mu}$ parameter (defined in Algorithm 1) for 10 epochs and fine-tune the $\boldsymbol{W}_{\text{stochastic}}$ parameter in the remaining 10 epochs, keeping $\boldsymbol{\mu}$ frozen. We refer to this experiment as 'post-hoc' `MonteCLoRA`. Decoupling the training of $\boldsymbol{\mu}$ and $\boldsymbol{W}_{\text{stochastic}}$ has several computational advantages. Post-hoc execution is particularly useful when the LLM is already fine-tuned and requires incremental calibration. On the

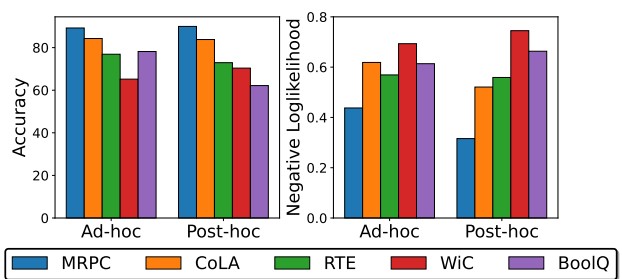

Figure 10: Effectiveness of `MonteCLoRA` in post-hoc operation.

downside, post-hoc execution might require more careful consideration, as if the fine-tuned model is struck in local optima, it may not be able to come out even after the post-hoc execution. Our observation with post-hoc `MonteCLoRA` in Figure 10 is rather more intriguing, where we observe performance improvement on smaller tasks – WiC (5%) and MRPC (0.7%) and almost similar performance on CoLA and RTE. On BoolQ, RoBERTa trained with `MonteCLoRA` post-hoc diverges due to instability during previous LoRA fine-tuning. However, it is important to note that existing Bayesian post-hoc operations require significant effort to mitigate the calibration and robustness challenges of low-rank fine-tuning. On the other hand, `MonteCLoRA` offers faster convergence (more discussion on this topic in the next section) with more robust performance, irrespective of whether it is applied ad-hoc or post-hoc.

## 7   Discussions

### 7.1   Convergence Analysis of `MonteCLoRA`

The previous section describes the sensitivity with existing low-rank adaptation-based fine-tuning methods. We also highlighted that given inappropriate hyperparameter selection, the existing techniques cannot demonstrate generalization capabilities post-fine-tuning process, defying the purpose of acquiring new knowledge during fine-tuning. In this section, we shed light on the training instability with these existing fine-tuning methods with two selected use cases.

Figure 11(a) illustrates a scenario where a LoRA fine-tuned RoBERTa-base model on the BoolQ task diverges. The training loss remains the same over the entire training period. Under the same hyperparameter configuration, `MonteCLoRA` achieves convergence with a steady drop in training loss. The training loss curve highlights that due to the posterior estimation, `MonteCLoRA` does not converge to any saddle point but rather robustly learns the global optima. A similar observation is made in Figure 11(b), where we observe abrasive optimization with LoRA fine-tuning strategy. RoBERTa-base with LoRA converges only after 6000 training steps. On the other hand, `MonteCLoRA` converges significantly faster in just $< 3000$ steps with a smoother optimization trajectory. With appropriate prior regularization, `MonteCLoRA` diminishes the sensitivity of the optimization process on hyperparameters like learning rates. Typically, with a high learning rate, a model can get struck at a local minima or even diverge, which is prominently seen with LoRA. However, this behavior is less frequent with `MonteCLoRA`.

### 7.2   Computational Complexity of `MonteCLoRA`

In Section 4, we discussed the asymptotic complexity of `MonteCLoRA` and highlighted that it introduces only $\mathcal{O}(r)$ additional parameters. While the vanilla implementation incurs runtime overhead due to online sampling of stochastic low-rank matrices during the forward pass, this overhead can be significantly reduced

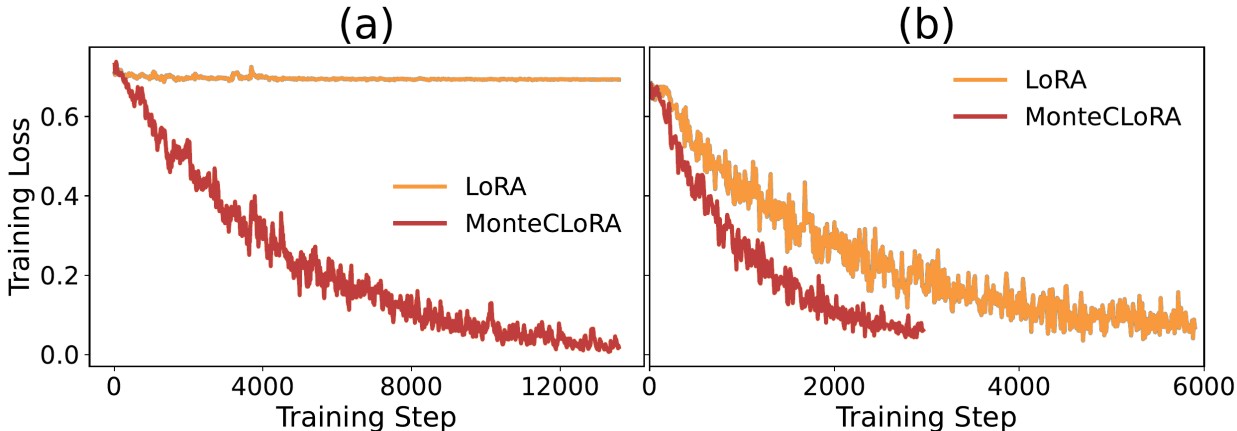

Figure 11: **(a)** Training loss curves on BoolQ with LoRA and `MonteCLoRA` fine-tuning. The loss curve with LoRA highlights its the instability during fine-tuning, where the model eventually diverges. **(b)** Training loss curves on WiC, where `MonteCLoRA` achieves twice faster convergence than LoRA. For both the cases, we use RoBERTa-base with a learning rate of $3 \times 10^{-4}$ and a batch size of 8 for both the fine-tuning strategies.

| Model | Modules | Runtime ($\times$ **LoRA**) | Memory ($\times$ **LoRA**) |
|---|---|---|---|
| RoBERTa-base | QKV | $1.63\times$ | $1.10\times$ |
| | All | $1.75\times$ | $1.15\times$ |
| LLaMA-3.2-3B-Instruct | QKV | $1.45\times$ | $1.07\times$ |
| | All | $1.78\times$ | $1.25\times$ |
| LLaMA-3.1-8B-Instruct | QKV | $1.23\times$ | $1.06\times$ |
| | All | $1.53\times$ | $1.23\times$ |
| LLaMA-2-13B | QKV | $1.27\times$ | $1.09\times$ |
| | All | $1.47\times$ | $1.13\times$ |

Table 12: Empirical runtime and memory overhead of `MonteCLoRA` compared to LoRA with buffered pre-sampling.

with appropriate buffered sampling. Specifically, we decouple the sampling from the forward pass by pre-generating samples and caching them in a fixed-size buffer. During training, samples are retrieved from this buffer via constant-time ($\mathcal{O}(1)$) lookups, effectively removing sampling latency from the critical path. We replenish the buffer once we have used all the samples. Since the sampling can be parallelized on GPU, the wall-clock time for replenishing the buffer is minimal. To evaluate this improved implementation, we benchmarked `MonteCLoRA` on the HumanEval dataset using LLaMA-3.2-3B-Instruct, LLaMA-3.1-8B-Instruct, and LLaMA-2-13B, and on the CoLA dataset using RoBERTa-base. For LLaMA models, we trained for 1000 training steps with an effective batch size of 8, a maximum sequence length of 512 tokens, and a LoRA rank of 32. For RoBERTa-base, we used a sequence length of 128 with a LoRA rank of 8. We experimented on combination query, key, and value matrices for both models. For RoBERTa-base, we also experimented with the dense matrix, and for LLaMA models, we experimented with output projection. For both models, we used a buffer size of 15. All our experiments were carried out on a single NVIDIA 80GB A100 GPU. The benchmarking results are presented in Table 12.

We observe that the runtime overhead of `MonteCLoRA` decreases significantly for larger models. For LLaMA-3.1-8B-Instruct, the relative training time is within 1.23–1.30$\times$ of LoRA, while memory usage remains under 1.10$\times$. This improvement stems from amortizing the cost of sampling over multiple forward passes. Moreover, in larger models, the forward pass dominates the overall training cost, making the sampling cost proportionally smaller. Overall, these results demonstrate that `MonteCLoRA` achieves a favorable trade-off between expressivity and efficiency, making it practical even for large-scale fine-tuning settings with long sequences and large models.

### 7.3 Limitations

While `MonteCLoRA` offers significant improvements in stability and robustness over prior low-rank fine-tuning methods, several limitations remain:

- **Sampling complexity:** The Monte Carlo estimation introduces additional training-time overhead. Although we have shown that this effect is smaller in larger models, it still increases training times.

- **GPU-dependent sampling latency:** Sampling from Wishart and multivariate Gaussian distributions can introduce variability in training time depending on the GPU hardware, memory bandwidth, and compute capabilities.

- **Additional hyperparameters:** `MonteCLoRA` introduces new hyperparameters such as the sample scaler $\epsilon$ and KL loss weight $\eta$ which may require tuning for each downstream task and model architecture.

- **Scale limitations:** `MonteCLoRA` has currently been evaluated on models of sizes up to 7B. Its scalability and effectiveness for larger models such as LLaMA-70B or Qwen-14B remain untested.

- **Synchronous sampling:** The sampling operations occur at regular intervals during the forward pass, introducing latency that could be mitigated with parallel or asynchronous methods.

### 7.4 Future Work

We identify several promising directions to extend this work:

- **Asynchronous sampling:** Designing an asynchronous sampler that decouples the sampling of Monte Carlo weights from the forward pass could significantly reduce latency and improve throughput.

- **Scaling to larger models:** Future work could explore fine-tuning of larger LLMs, such as LLaMA-70B to test `MonteCLoRA`'s robustness at scale and in longer-context settings.

- **Adaptive sample scaler scheduling:** Rather than fixing the sample scaler, an adaptive scheduling strategy based on layer-wise uncertainty, entropy or the variation in cross entropy loss could improve learning and generalization.

- **Variational inference extensions:** Replacing the Dirichlet-Wishart sampling with variational approximations may yield tighter bounds and faster convergence.

- **Inference-time posterior averaging:** Exploring lightweight ensemble techniques using posterior samples could improve prediction calibration and performance in out-of-distribution settings during test time.

- **Cross-domain extensions:** Adapting `MonteCLoRA` to vision transformers, speech models, or multimodal architectures could extend its applicability beyond language.

## 8 Conclusion

This paper introduced a Bayesian reparameterization of low-rank adaptation for efficiently fine-tuning LLMs. We highlighted the sensitivity of existing fine-tuning strategies and proposed `MonteCLoRA` that provides a robust and unbiased estimate of parameter-efficient fine-tuning. `MonteCLoRA` parameterizes low-rank fine-tuned parameters as a mixture of Gaussian with appropriate prior parameterization. With robust Monte Carlo estimates, `MonteCLoRA` reduces the sensitivity of the parameterized LLMs over the hyperparameters. Our thorough empirical study overmined the superiority of `MonteCLoRA` over the contemporary low-rank adaptation method in terms of performance and stability. Although the current work acknowledges the effectiveness of `MonteCLoRA` on low-rank parameterization regimes, the applicability of our method extends

far beyond this. `MonteCLoRA` can be equally effective with the pre-trained LLM parameters. However, keeping the rising computation cost of fine-tuning LLMs, it is computationally more sensible to utilize `MonteCLoRA` with low-rank parameters rather than high-rank dense parameters. `MonteCLoRA` can also offer great flexibility where it can be used to ensure robustness even during pre-training of LLMs.

## Acknowledgment

T. Chakraborty acknowledges the support of the J.P. Morgan AI Faculty Research Award and Rajiv Khemani Young Faculty Chair Professorship in Artificial Intelligence. This paper was prepared for informational purposes in part by the Artificial Intelligence Research group of JPMorgan Chase & Co and its affiliates ("J.P. Morgan") and is not a product of the Research Department of J.P. Morgan. J.P. Morgan makes no representation and warranty whatsoever and disclaims all liability, for the completeness, accuracy or reliability of the information contained herein. This document is not intended as investment research or investment advice, or a recommendation, offer or solicitation for the purchase or sale of any security, financial instrument, financial product or service, or to be used in any way for evaluating the merits of participating in any transaction, and shall not constitute a solicitation under any jurisdiction or to any person, if such solicitation under such jurisdiction or to such person would be unlawful.

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

# A    Additional Proofs

## A.1    Proof of Lemma 3.1

Let $\boldsymbol{X}, \boldsymbol{Y} \in \mathbb{R}^n$ be arbitrary points. By the Mean Value Theorem for vector-valued functions, there exists a point $\boldsymbol{\xi}$ lying on the line segment connecting $\boldsymbol{X}$ and $\boldsymbol{Y}$ such that

$$\nabla J(\boldsymbol{X}) - \nabla J(\boldsymbol{Y}) = H(\boldsymbol{\xi})(\boldsymbol{X} - \boldsymbol{Y}). \tag{15}$$

Taking the norm on both sides, we have,

$$\|\nabla J(\boldsymbol{X}) - \nabla J(\boldsymbol{Y})\| = \|H(\boldsymbol{\xi})(\boldsymbol{X} - \boldsymbol{Y})\|. \tag{16}$$

Using the submultiplicative property of norms, specifically the operator norm for matrices and the Euclidean norm for vectors, we obtain

$$\|H(\boldsymbol{\xi})(\boldsymbol{X} - \boldsymbol{Y})\| \leq \|H(\boldsymbol{\xi})\| \cdot \|\boldsymbol{X} - \boldsymbol{Y}\|, \tag{17}$$

where $\|H(\boldsymbol{\xi})\|$ denotes the operator norm (spectral norm) of the Hessian matrix $H(\boldsymbol{\xi})$. Since $H(\boldsymbol{\xi})$ is symmetric (because $J$ is twice continuously differentiable), its operator norm equals its largest eigenvalue in absolute value

$$\|H(\boldsymbol{\xi})\| = \lambda_{\max}(H(\boldsymbol{\xi})). \tag{18}$$

Therefore, we have,

$$\|\nabla J(\boldsymbol{X}) - \nabla J(\boldsymbol{Y})\| \leq \lambda_{\max}(H(\boldsymbol{\xi})) \cdot \|\boldsymbol{X} - \boldsymbol{Y}\|. \tag{19}$$

Since $\boldsymbol{\xi}$ lies on the line segment between $\boldsymbol{X}$ and $\boldsymbol{Y}$, and $\boldsymbol{X}$ and $\boldsymbol{Y}$ are arbitrary in $\mathbb{R}^n$, we can take the supremum over all such $\boldsymbol{\xi}$

$$\|\nabla J(\boldsymbol{X}) - \nabla J(\boldsymbol{Y})\| \leq \left( \sup_{\boldsymbol{\xi} \in \mathbb{R}^n} \lambda_{\max}(H(\boldsymbol{\xi})) \right) \|\boldsymbol{X} - \boldsymbol{Y}\|. \tag{20}$$

Thus, the Lipschitz constant $L$ of the gradient $\nabla J$ is given by

$$L = \sup_{\boldsymbol{X} \in \mathbb{R}^n} \lambda_{\max}(H(\boldsymbol{X})). \tag{21}$$

## A.2    Proof of Lemma 3.2

The smoothed loss function $\tilde{J}(\boldsymbol{\theta})$ can be expanded as,

$$\tilde{J}(\boldsymbol{\theta}) = \int J(\boldsymbol{\theta} + \boldsymbol{\gamma})\mathcal{N}(\boldsymbol{\gamma}; 0, \boldsymbol{\Sigma})d\boldsymbol{\gamma}.$$

The gradient of $\tilde{J}(\boldsymbol{\theta})$ is,

$$\nabla_{\boldsymbol{\theta}}\tilde{J}(\boldsymbol{\theta}) = \int \nabla_{\boldsymbol{\theta}} J(\boldsymbol{\theta} + \boldsymbol{\gamma})\mathcal{N}(\boldsymbol{\gamma}; 0, \boldsymbol{\Sigma})d\boldsymbol{\gamma}.$$

The Hessian of $\tilde{J}(\boldsymbol{\theta})$ is,

$$\tilde{H}(\boldsymbol{\theta}) = \nabla_{\boldsymbol{\theta}}^2 \tilde{J}(\boldsymbol{\theta}) = \int \nabla_{\boldsymbol{\theta}}^2 L(\boldsymbol{\theta} + \boldsymbol{\gamma})\mathcal{N}(\boldsymbol{\gamma}; 0, \boldsymbol{\Sigma})d\boldsymbol{\gamma}.$$

So, the Hessian is the expectation of the Hessian at $\boldsymbol{\theta}$, i.e.,

$$\tilde{H}(\boldsymbol{\theta}) = \mathbb{E}_{\boldsymbol{\gamma}}[H(\boldsymbol{\theta} + \boldsymbol{\gamma})].$$

Since $H(\boldsymbol{\theta})$ is symmetric and positive semi-definite (for convex $L$), so is $H(\boldsymbol{\theta} + \boldsymbol{\gamma})$ for all $\boldsymbol{\gamma}$. The expected Hessian $\tilde{H}(\boldsymbol{\theta})$ is a convex combination (integral) of Hessians at points $\boldsymbol{\theta} + \boldsymbol{\gamma}$. Therefore, $\tilde{H}(\boldsymbol{\theta})$ is also symmetric and positive semi-definite. Let $v$ be an eigenvector of $\tilde{H}(\boldsymbol{\theta})$ with eigenvalue $\lambda$, then,

$$\tilde{H}(\boldsymbol{\theta})\boldsymbol{v} = \lambda\boldsymbol{v}.$$

Using the fact that $\boldsymbol{v}^T\boldsymbol{v}$ is a scalar, we have,

$$\lambda = \frac{\boldsymbol{v}^\top \tilde{H}(\boldsymbol{\theta})\boldsymbol{v}}{\boldsymbol{v}^\top \boldsymbol{v}} = \frac{\boldsymbol{v}^\top \mathbb{E}_{\boldsymbol{\gamma}}[H(\boldsymbol{\theta} + \boldsymbol{\gamma})]\boldsymbol{v}}{\boldsymbol{v}^\top \boldsymbol{v}}.$$

Since $\mathbb{E}$ is linear, we have,

$$\lambda = \mathbb{E}_{\boldsymbol{\gamma}}\left[\frac{\boldsymbol{v}^\top H(\boldsymbol{\theta} + \boldsymbol{\gamma})\boldsymbol{v}}{\boldsymbol{v}^\top \boldsymbol{v}}\right].$$

Let us define

$$S(\boldsymbol{\theta} + \boldsymbol{\gamma}) = \frac{\boldsymbol{v}^\top H(\boldsymbol{\theta} + \boldsymbol{\gamma})\boldsymbol{v}}{\boldsymbol{v}^\top \boldsymbol{v}}.$$

Therefore,

$$\lambda = \mathbb{E}_{\boldsymbol{\gamma}}[S(\boldsymbol{\theta} + \boldsymbol{\gamma})].$$

Since $H(\boldsymbol{\theta} + \boldsymbol{\gamma})$ is positive semi-definite, its eigenvalues $S(\boldsymbol{\theta} + \boldsymbol{\gamma}) \geq 0$. Let $\Lambda_{\max}$ be the maximum eigenvalue of $H(\boldsymbol{\theta})$ over all $\boldsymbol{\theta}$. Then, for all $\boldsymbol{\gamma}$

$$S(\boldsymbol{\theta} + \boldsymbol{\gamma}) \leq \Lambda_{\max}.$$

Taking expectation we get,

$$\lambda = \mathbb{E}_{\boldsymbol{\gamma}}[S(\boldsymbol{\theta} + \boldsymbol{\gamma})] \leq \Lambda_{\max}.$$

Hence,

$$\lambda_{\max}(\tilde{H}(\boldsymbol{\theta})) = \sup_{\boldsymbol{v} \neq 0} \frac{\boldsymbol{v}^\top \tilde{H}(\boldsymbol{\theta})\boldsymbol{v}}{\boldsymbol{v}^\top \boldsymbol{v}} \leq \Lambda_{\max}.$$

# B   Datasets

## B.1   Natural Language Understanding

The General Language Understanding Evaluation (GLUE) (Wang et al., 2018) and SuperGLUE benchmarks (Wang et al., 2019) evaluate the language understanding capabilities of LLMs. All the datasets in GLUE and SuperGLUE are obtained from `https://gluebenchmark.com/` and `https://super.gluebenchmark.com/`, respectively. We elaborate on the GLUE and SuperGLUE tasks as follows:

**CoLA** (Warstadt et al., 2019), or The Corpus of Linguistic Acceptability, comprises acceptability judgments from books and journal articles. The task is to indicate the grammatical correctness of the given sentence as "acceptable" or "unacceptable" using the Matthews correlation coefficient as the evaluation metric.

**MRPC** (Dolan & Brockett, 2005), or Microsoft Research Paraphrase Corpus, comprises pairs of sentences extracted from online news sources. The objective is to predict whether the provided pair of sentences are paraphrases of each other or not.

**RTE** (Dagan et al., 2005; Haim et al., 2006; Giampiccolo et al., 2007; Bentivogli et al., 2009), or Recognizing Textual Entailment, are formed by combining a series of annual textual entailment challenges. The task comprises categorizing whether the two sentences entail each other.

**BoolQ** (Clark et al., 2019), or Boolean Questions, involves binary inquiries sourced from the Google search engine. These questions are combined with pertinent paragraphs extracted from Wikipedia articles, ensuring that the provided paragraphs contain accurate answers to the queries.

**WiC** (Pilehvar & Camacho-Collados, 2018), or Word-in-Context, is a word sense disambiguation task that involves binary classification of sentence pairs. Within this task, two text snippets are presented, each featuring a word with multiple potential meanings. The objective is to determine whether the specified word holds the same meaning in both sentences.

### B.2 Commonsense Reasoning

**PiQA** (Bisk et al., 2019), or Physical Interaction: Question Answering, is a reasoning benchmark based on physical interactions in everyday situations. This benchmark comprises questions with two choices, out of which one option is correct based on realistic understanding of the world.

**SiQA** (Sap et al., 2019) or Social Interaction Question Answer, is a reasoning benchmark based on testing social commonsense intelligence. SiQA consists of multiple choice questions, with each question describing a social scenario, followed by a query with multiple options, one of which is the correct scenario representing the protagonist's likely intention.

**Winogrande** (Sakaguchi et al., 2021) is a commonsense reasoning task inspired by the WSC (Levesque et al., 2012). The goal of the task is to choose the correct choice given two choices.

**ARC-easy and ARC-challenge** (Clark et al., 2018) comprise the AI2 Reasoning Challenge (ARC) partitioned into easy (ARC-e) and challenging set (ARC-c). The datasets consist of grade-school science questions in multiple choice format.

**OpenBookQA** (Mihaylov et al., 2018) contains elementary-level science questions in the form of multiple choices requiring additional commonsense knowledge.

### B.3 GSM8k and Humaneval

**GSM8k** (Cobbe et al., 2021) is a benchmark designed to evaluate arithmetic and mathematical reasoning in a step-by-step setting. It consists of grade-school level word problems requiring multi-step numerical reasoning. We use the official test split to evaluate model performance on this task. Following prior work, we report accuracy using the *exact match* (EM) metric, which measures the proportion of generated answers that exactly match the reference answer.

To account for variations in output formatting and answer representation, we apply two post-processing filters on model generations: *strict-match* and *flexible-extract*. The strict-match filter expects the model's final output to contain only the correct answer without any surrounding context. In contrast, the flexible-extract filter employs regular expressions to parse and extract the final answer from longer outputs that may include chains of thought, explanations, or extra tokens. Reporting both versions allows us to separately quantify a model's reasoning accuracy and its ability to follow formatting instructions precisely.

**HumanEval** (Chen et al., 2021) is a widely used benchmark for evaluating code generation through functional correctness. Each problem in HumanEval consists of a prompt and hidden unit tests that validate correctness. We evaluate using the *pass@k* metric, where $k$ denotes the number of independently sampled completions per problem. A problem is considered passed if at least one of the $k$ completions passes all unit tests. We report results for $k = 2$ and $k = 4$, following standard protocol. The final pass@k score corresponds to the fraction of problems in the dataset for which at least one generated solution is functionally correct.

To create a lightweight yet representative training set for code generation, we curate a subset from the Magicoder-OSS-Instruct-75K corpus (Wei et al., 2024). We filter for Python problems whose tokenized input length does not exceed 512 tokens and randomly sample 10,000 such instances.

## C Distributions and Reparameterization for Differentiable Backpropagation

Reparameterization trick (Kingma, 2013) is often used to make sampling from a probability distribution differentiable, where the distribution parameters are unknown and learnable parameters of the model. Our method is heavily dependent on reparameterized sampling from different distributions, their correctness and their differentiability. Here we describe the reparameterization techniques for the different probability distributions used in the paper.

### C.1 Multivariate Gaussian

We denote a normally distributed random vector as $\mathbf{x} \sim \mathcal{N}(\boldsymbol{\mu}, \boldsymbol{\Sigma})$. The probability density function of the Normal distribution is given by

$$f_{\mathbf{x}}(\mathbf{x}) = \frac{1}{(2\pi)^{p/2}|\boldsymbol{\Sigma}|^{1/2}} \exp\left(-\frac{1}{2}(\mathbf{x} - \boldsymbol{\mu})^{\top}\boldsymbol{\Sigma}^{-1}(\mathbf{x} - \boldsymbol{\mu})\right),$$

where,

- $(\mathbf{x} - \boldsymbol{\mu})$ is the deviation of the random vector from the mean.

- $\boldsymbol{\Sigma}^{-1}$ is the inverse of the covariance matrix.

- $|\boldsymbol{\Sigma}|$ denotes the determinant of $\boldsymbol{\Sigma}$.

**Reparameterized Sampling and Proof of Differentiability**

Let $\boldsymbol{x} \sim \mathcal{N}(\boldsymbol{\mu}, \boldsymbol{\Sigma})$ be a random vector sampled from a multivariate normal distribution, where $\boldsymbol{\mu} \in \mathbb{R}^d$ is the mean vector, and $\boldsymbol{\Sigma} \in \mathbb{R}^{d \times d}$ is the covariance matrix. In the reparameterization trick, we express the sample $\boldsymbol{x}$ as,

$$\boldsymbol{x} = \boldsymbol{\mu} + \boldsymbol{L}\boldsymbol{\epsilon},$$

where $\boldsymbol{\epsilon} \sim \mathcal{N}(\boldsymbol{0}, \boldsymbol{I})$ is a standard normal vector, and $\boldsymbol{L}$ is the Cholesky decomposition of the covariance matrix $\boldsymbol{\Sigma}$, such that $\boldsymbol{\Sigma} = \boldsymbol{L}\boldsymbol{L}^T$. We will prove that the reparameterized transformation $\boldsymbol{x} = \boldsymbol{\mu} + \boldsymbol{L}\boldsymbol{\epsilon}$ is differentiable with respect to both $\boldsymbol{\mu}$ and $\boldsymbol{\Sigma}$. The mean vector $\boldsymbol{\mu}$ appears linearly in the transformation

$$\boldsymbol{x} = \boldsymbol{\mu} + \boldsymbol{L}\boldsymbol{\epsilon}.$$

Since the transformation is linear with respect to $\boldsymbol{\mu}$, the derivative of $\boldsymbol{x}$ with respect to $\boldsymbol{\mu}$ is,

$$\frac{\partial \boldsymbol{x}}{\partial \boldsymbol{\mu}} = \boldsymbol{I},$$

where $\boldsymbol{I}$ is the identity matrix. Hence, $\boldsymbol{x}$ is differentiable with respect to $\boldsymbol{\mu}$.

The covariance matrix $\boldsymbol{\Sigma}$ enters the transformation through Cholesky decomposition $\boldsymbol{\Sigma} = \boldsymbol{L}\boldsymbol{L}^T$. The Cholesky decomposition is a differentiable function of $\boldsymbol{\Sigma}$ as long as $\boldsymbol{\Sigma}$ is positive definite. Thus, the entries of the matrix $\boldsymbol{L}$ are differentiable functions of the entries of $\boldsymbol{\Sigma}$. Since the transformation $\boldsymbol{x} = \boldsymbol{\mu} + \boldsymbol{L}\boldsymbol{\epsilon}$ depends linearly on $\boldsymbol{L}$, we can apply the chain rule to differentiate $\boldsymbol{x}$ with respect to $\boldsymbol{\Sigma}$. Let $\boldsymbol{x} = f(\boldsymbol{\Sigma}, \boldsymbol{\epsilon}) = \boldsymbol{\mu} + g(\boldsymbol{\Sigma})\boldsymbol{\epsilon}$, where $g(\boldsymbol{\Sigma})$ is the Cholesky factor $\boldsymbol{L}$. The derivative of $\boldsymbol{x}$ with respect to $\boldsymbol{\Sigma}$ is

$$\frac{\partial \boldsymbol{x}}{\partial \boldsymbol{\Sigma}} = \frac{\partial \boldsymbol{x}}{\partial \boldsymbol{L}} \cdot \frac{\partial \boldsymbol{L}}{\partial \boldsymbol{\Sigma}}.$$

Since $\boldsymbol{x}$ is a linear transformation of $\boldsymbol{L}$, and $\boldsymbol{L}$ is differentiable with respect to $\boldsymbol{\Sigma}$, $\boldsymbol{x}$ is differentiable with respect to $\boldsymbol{\Sigma}$.

### C.2 Wishart Distribution

We denote a Wishart-distributed random matrix as $\mathbf{W} \sim \mathcal{W}_p(\mathbf{V}, \nu)$. The probability density function of the Wishart distribution is given by,

$$f_{\mathbf{W}}(\mathbf{W}) = \frac{|\mathbf{W}|^{(\nu - p - 1)/2} \exp\left(-\frac{1}{2}\operatorname{tr}\left(\mathbf{V}^{-1}\mathbf{W}\right)\right)}{2^{\nu p/2}|\mathbf{V}|^{\nu/2}\Gamma_p\left(\frac{\nu}{2}\right)}.$$

**Reparameterized Sampling and Proof of Differentiability**

Wishart distribution is the conjugate prior of the precision matrix (inverse covariance-matrix) of a multivariate Gaussian distribution with unknown variance. For sampling $\boldsymbol{\Sigma} \sim \mathcal{W}_m(\boldsymbol{V}, n)$ with $\boldsymbol{V} \in \mathbb{R}^{m \times m}$ being the

scale matrix and $n$ degrees of freedom, we first calculate Cholesky decomposition $\boldsymbol{LL}^T = \boldsymbol{V}$, with a lower triangular matrix $\boldsymbol{L}$. As $\boldsymbol{V}$ is diagonal, $\boldsymbol{L}$ can be directly calculated as

$$
\begin{bmatrix}
\sqrt{V_1} & & \\
& \ddots & \\
& & \sqrt{V_m}
\end{bmatrix}.
$$

Next, we sample $\tilde{\boldsymbol{\Sigma}} \sim \mathcal{W}_m(\boldsymbol{I}, n)$. The reparametrized variance matrix is $\boldsymbol{\Sigma} = \boldsymbol{L}\tilde{\boldsymbol{\Sigma}}\boldsymbol{L}^T$. Let $\boldsymbol{\Sigma} \sim \mathcal{W}_p(\boldsymbol{V}, n)$ be a sample from the Wishart distribution, where $\boldsymbol{V} \in \mathbb{R}^{p \times p}$ is a positive-definite scale matrix, and $n$ is the degrees of freedom.

The Cholesky decomposition relates $\boldsymbol{V}$ and $\boldsymbol{L}$ through the equation $\boldsymbol{V} = \boldsymbol{LL}^T$. The elements of $\boldsymbol{V}$ are quadratic functions of the elements of $\boldsymbol{L}$,

$$
V_{ij} = \sum_{k=1}^{\min(i,j)} \ell_{ik}\ell_{jk}.
$$

This relationship is differentiable, and the partial derivatives of $\ell_{ij}$ with respect to $V_{kl}$ are well-defined. Thus, $\boldsymbol{L}$ is a differentiable function of $\boldsymbol{V}$. The reparameterized Wishart sample is given by,

$$
\boldsymbol{\Sigma} = \boldsymbol{L}\tilde{\boldsymbol{\Sigma}}\boldsymbol{L}^T.
$$

Using the chain rule, we compute the derivative of $\boldsymbol{\Sigma}$ with respect to $\boldsymbol{V}$ as,

$$
\frac{\partial \boldsymbol{\Sigma}}{\partial \boldsymbol{V}} = \frac{\partial \boldsymbol{\Sigma}}{\partial \boldsymbol{L}} \cdot \frac{\partial \boldsymbol{L}}{\partial \boldsymbol{V}}.
$$

The differentiation of $\boldsymbol{\Sigma}$ with respect to $\boldsymbol{V}$ proceeds as follows. Using the chain rule, we first differentiate $\boldsymbol{\Sigma}$ with respect to $\boldsymbol{L}$, and then differentiate $\boldsymbol{L}$ with respect to $\boldsymbol{V}$

$$
\frac{\partial \boldsymbol{\Sigma}}{\partial \boldsymbol{V}} = \frac{\partial \boldsymbol{\Sigma}}{\partial \boldsymbol{L}} \cdot \frac{\partial \boldsymbol{L}}{\partial \boldsymbol{V}}.
$$

Since $\boldsymbol{\Sigma} = \boldsymbol{L}\tilde{\boldsymbol{\Sigma}}\boldsymbol{L}^T$, the derivative with respect to $\boldsymbol{L}$ is,

$$
\frac{\partial \boldsymbol{\Sigma}}{\partial \boldsymbol{L}} = \frac{\partial}{\partial \boldsymbol{L}} \left( \boldsymbol{L}\tilde{\boldsymbol{\Sigma}}\boldsymbol{L}^T \right) = \tilde{\boldsymbol{\Sigma}}\boldsymbol{L}^T + \boldsymbol{L}\tilde{\boldsymbol{\Sigma}}.
$$

From the above, we can conclude that the reparamterised fromulation of Wishart Distribution retains its differentiability with respect to its parameters

### C.3 Dirichlet Distribution

We denote a Dirichlet-distributed random vector as $\boldsymbol{\theta} \sim \text{Dir}(\boldsymbol{\alpha})$, where $\boldsymbol{\theta} = (\theta_1, \theta_2, \ldots, \theta_K)$ and $\sum_{i=1}^{K} \theta_i = 1$. The probability density function of the Dirichlet distribution is given by,

$$
f_{\boldsymbol{\theta}}(\boldsymbol{\theta}) = \frac{1}{B(\boldsymbol{\alpha})} \prod_{i=1}^{K} \theta_i^{\alpha_i - 1},
$$

where

- $\theta_i \geq 0$ for all $i = 1, \ldots, K$, and $\sum_{i=1}^{K} \theta_i = 1$.

- $B(\boldsymbol{\alpha})$ is the multivariate beta function, defined as

$$
B(\boldsymbol{\alpha}) = \frac{\prod_{i=1}^{K} \Gamma(\alpha_i)}{\Gamma\left(\sum_{i=1}^{K} \alpha_i\right)}
$$

- $\Gamma(\cdot)$ is the Gamma function.

**Reparameterized Sampling and Proof of Differentiability**
Suppose we want to sample $\boldsymbol{\pi} \sim \text{Dir}(\boldsymbol{\alpha})$, where $\boldsymbol{\alpha} = (\alpha_1, \ldots, \alpha_K)$ is the concentration parameter. We first sample $K$ independent Gamma-distributed variables $y_i \sim \text{Gamma}(\alpha_i, 1)$, for $i = 1, \ldots, K$. To ensure differentiability of the sampling process, we apply the reparameterization trick by expressing each $y_i$ as,

$$y_i = \alpha_i \cdot (-\log \epsilon_i),$$

where $\epsilon_i \sim \text{Uniform}(0, 1)$ is an independent uniform random variable. Finally, the reparameterized Dirichlet sample $\boldsymbol{p}$ is obtained as follows.

We sample $K$ independent Gamma-distributed variables $y_i \sim \text{Gamma}(\alpha_i, 1)$, for $i = 1, \ldots, K$. The reparameterization of the Gamma distribution is given by,

$$y_i = g(\alpha_i, \epsilon_i) = \alpha_i \cdot (-\log(\epsilon_i)),$$

where $\epsilon_i \sim \text{Uniform}(0, 1)$. Since the transformation $g(\alpha_i, \epsilon_i)$ is differentiable with respect to $\alpha_i$, we have,

$$\frac{\partial y_i}{\partial \alpha_i} = -\log(\epsilon_i).$$

Thus, $y_i$ is differentiable with respect to $\alpha_i$. After sampling $y_i$, we normalize the variables to obtain the Dirichlet sample,

$$p_i = \frac{y_i}{\sum_{j=1}^{K} y_j}.$$

The normalization is a smooth function of the $y_i$s, so the resulting $p_i$ is differentiable as long as $\sum_{j=1}^{K} y_j > 0$ (which holds because each $y_i > 0$). The derivative of $p_i$ with respect to $y_i$ is,

$$\frac{\partial p_i}{\partial y_i} = \frac{\sum_{j=1}^{K} y_j - y_i}{\left(\sum_{j=1}^{K} y_j\right)^2}.$$

Since $y_i$ is differentiable with respect to $\alpha_i$, the sample $\boldsymbol{p} = \{p_1, p_2, \cdots, p_K\}$ is differentiable with respect to $\boldsymbol{\alpha}$.

# D   Calculation of KL Divergence Losses

For calculating the KL Divergence of the distributions used in this paper, we refer to Soch et al. (2024). The detailed derivations can also be found in Section 1 of the supplementary material.

**Multivariate Normal.**   The KL divergence between $P \sim \mathcal{N}(\boldsymbol{\mu_1}, \boldsymbol{\Sigma_1})$ and $Q \sim \mathcal{N}(\boldsymbol{\mu_2}, \boldsymbol{\Sigma_2})$ is given by,

$$KL[P||Q] = \frac{1}{2}\left[(\boldsymbol{\mu_2} - \boldsymbol{\mu_1})^T \boldsymbol{\Sigma_2}^{-1}(\boldsymbol{\mu_2} - \boldsymbol{\mu_1}) + \text{tr}(\boldsymbol{\Sigma_2}^{-1}\boldsymbol{\Sigma_1}) - \ln\frac{|\boldsymbol{\Sigma_1}|}{|\boldsymbol{\Sigma_2}|} - n\right].$$

**Wishart Distribution.**   The KL divergence between $P \sim \mathcal{W}_p(\mathbf{V_1}, n_1)$ and $Q \sim \mathcal{W}_p(\mathbf{V_2}, n_2)$ is given by,

$$KL[P||Q] = \frac{1}{2}\left[n_2\left(\ln|\boldsymbol{V_2}| - \ln|\boldsymbol{V_1}|\right) + n_1\text{tr}(\boldsymbol{V_2}^{-1}\boldsymbol{V_1}) + 2\ln\frac{\Gamma_p\left(\frac{n_2}{2}\right)}{\Gamma_p\left(\frac{n_1}{2}\right)} + (n_1 - n_2)\psi_p\left(\frac{n_1}{2}\right) - n_1 p\right].$$

**Dirichlet Distribution.**   The KL divergence between $P \sim \text{Dir}(\alpha_1)$ from $Q \sim \text{Dir}(\alpha_2)$ is given by,

$$KL[P||Q] = \ln\frac{\Gamma\left(\sum_{i=1}^{k}\alpha_{1,i}\right)}{\Gamma\left(\sum_{i=1}^{k}\alpha_{2,i}\right)} + \sum_{i=1}^{k}\ln\frac{\Gamma(\alpha_{2,i})}{\Gamma(\alpha_{1,i})} + \sum_{i=1}^{k}(\alpha_{1,i} - \alpha_{2,i})\left[\psi(\alpha_{1,i}) - \psi\left(\sum_{i=1}^{k}\alpha_{1,i}\right)\right].$$

# E Complexity Analysis of Sampling

Training `MonteCLoRA` involves sampling from three different distributions. Therefore, it is essential to analyze the complexity of sampling from these distributions.

## E.1 Sampling from the Wishart Distribution

1. **Cholesky Decomposition of the Scale Matrix.** The time complexity of computing the Cholesky decomposition for a $p \times p$ matrix is typically $\mathcal{O}(p^3)$. However, since we are using a diagonal prior, the time complexity reduces to $\mathcal{O}(p)$ or $\mathcal{O}(r)$, as we are only sampling for LoRA $A$, and the prior is $r \times r$.

2. **Sampling from the Standard Wishart Distribution $\tilde{\boldsymbol{\Sigma}} \sim \mathcal{W}_p(\boldsymbol{I}, k)$.** We generate $k$ independent samples from a multivariate normal distribution $\mathcal{N}(\boldsymbol{0}, \boldsymbol{I}_p)$, with each sample involving $p$ values. Generating the covariance matrix for each sample takes $\mathcal{O}(p^2)$ time. Therefore, the total time complexity for generating $k$ samples is $\mathcal{O}(kp^2)$. In our context, this corresponds to $\mathcal{O}(n_{\text{in}}r^2)$.

3. **Reparameterization $\boldsymbol{\Sigma} = \boldsymbol{L}\tilde{\boldsymbol{\Sigma}}\boldsymbol{L}^T$.** The reparameterization step involves matrix multiplication and thus has a time complexity of $\mathcal{O}(p^3)$. In our case, it is $\mathcal{O}(r^3)$.

The total time complexity is $\mathcal{O}(r^3 + n_{\text{in}}r^2)$. Since $r \ll \min(n_{\text{in}}, n_{\text{out}})$, the complexity is heavily influenced by $n_{\text{in}}$. To mitigate this problem, we reduce the degrees of freedom to $\mathcal{O}(r)$. In practice, we found little difference between two degrees of freedom; therefore, we use the Wishart prior with degrees of freedom $r$.

## E.2 Sampling from the Dirichlet Distribution

The time complexity of sampling from the Dirichlet distribution using the reparameterization trick can be broken down into the following steps

1. **Reparameterization via Gamma Distribution.** Sampling from a Gamma distribution with shape parameter $\alpha_i$ and scale parameter 1 has time complexity $\mathcal{O}(1)$ for each variable $y_i$. Sampling $N$ independent Gamma-distributed variables takes $\mathcal{O}(N)$ time.

2. **Normalization Step.** After generating Gamma-distributed variables $y_i$, we normalize them to produce the Dirichlet-distributed sample $\boldsymbol{p} = (p_1, \ldots, p_N)$. Calculating the denominator involves a summation taking $\mathcal{O}(N)$ time, and normalizing each variable takes $\mathcal{O}(1)$ time. Therefore, the total time complexity for the normalization step is $\mathcal{O}(N)$.

Thus, the total time complexity of sampling from a Dirichlet distribution is $\mathcal{O}(N)$.

## E.3 Sampling from a Multivariate Normal Distribution

The time complexity of sampling from a multivariate normal distribution using the reparameterization trick is as follows

1. **Cholesky Decomposition of the Covariance Matrix.** The time complexity for computing the Cholesky decomposition of a $d \times d$ covariance matrix $\boldsymbol{\Sigma}$ is $\mathcal{O}(d^3)$. Since we use a diagonal covariance matrix, the complexity reduces to $\mathcal{O}(d)$. For us, it is $\mathcal{O}(r)$.

2. **Sampling from the Standard Normal Distribution.** The time complexity for generating a vector of $d$ standard normal variables is $\mathcal{O}(d)$. For our case, this is $\mathcal{O}(r)$.

3. **Reparameterization $\boldsymbol{x} = \boldsymbol{\mu} + \boldsymbol{L}\boldsymbol{\epsilon}$.** The matrix-vector multiplication $\boldsymbol{L}\boldsymbol{\epsilon}$ involves multiplying a $d \times d$ matrix by a $d \times 1$ vector, which has a time complexity of $\mathcal{O}(d^2)$. For `MonteCLoRA`, this is $\mathcal{O}(r^2)$.

Thus, sampling one random variable from this multivariate normal distribution has a time complexity of $\mathcal{O}(r^2)$. Since we need to sample $n_{\text{in}}$ random variables for a weight matrix, the total time complexity for sampling is $\mathcal{O}(n_{\text{in}}r^2)$.

### E.4 Total Sampling Complexity for the Entire Model

Combining the time complexities for the three sampling steps mentioned above, the time complexity for sampling in one layer is,

$$\mathcal{O}(r^3 + n_{\text{in}}r^2 + N + n_{\text{in}}r^2),$$

which simplifies to

$$\mathcal{O}(n_{\text{in}}r^2 + N),$$

assuming that $r \ll \max(n_{\text{in}}, n_{\text{out}})$. Let $M_{in}$ be the largest input dimension for any layer in the model, and let the number of `MonteCLoRA`-enhanced layers be $L_{mc}$. The total time complexity for sampling across all layers is,

$$\mathcal{O}(L_{mc}(n_{\text{in}}r^2 + N)).$$

Since $n_{\text{in}}$ is typically constant for most models, the total time complexity is primarily governed by the number of `MonteCLoRA` layers, $L_{mc}$, and the LoRA rank, $r$.

