# OpenReview forum: "Robust and Efficient Fine-tuning of LLMs with Bayesian Reparameterization of Low-Rank Adaptation"
_TMLR — Accepted by TMLR_

### Review · Reviewer_QNMP · 2025-04-29

**Summary Of Contributions:**

- The authors devise a method to improve the robustness/variance of the LoRA fine-tuning process
- The authors present evidence of MonteCLoRA's performance on the GLUE benchmark (ROBERTa) as well as various NLG tasks (Llama7b).

**Audience:**

Yes

**Claims And Evidence:**

Yes

**Requested Changes:**

- The way the introduction is worded may lead the reader to believe that the robustness of finetuning with LoRA is being addressed, as it is worse than that of full fine-tuning, which is not the case as shown in Fig. 1. I would like for MonteCLoRA to be presented in Fig. 1 as well and the motivation be a bit more clear.
- I would like to see a comparison with alternative PEFT methods such as IA3 and Prompt Tuning

**Strengths And Weaknesses:**

- The authors provide a theoretical argument for their proposed method's robustness and that the final model output by MonteCLoRA is an unbiased estimator of the original model output
- The authors do not provide experimental results on more common NLG benchmarks such as GSM8k and HumanEval.
- The proposed method is ~2× slower than LoRA

---

> ### Author Response · Authors · 2025-05-18
> **Acknowledgement of reviewer QNMP comments**
>
> Dear reviwerer QNMP,
>
> We appreciate your valuable comments on our paper. We are working on addressing your comments and will be providing the clarifications soon.
>
> Thanks,

---

> ### Author Response · Authors · 2025-06-15
> **Response to Reviewer QNMP Comments - part I**
>
> We thank reviewer QNMP for their valuable comments. We address the raised queries in the following responses. We have also made the necessary changes in our revised manuscript (highlighted in blue font).
>
> **W1: Evaluation on NLG benchmarks with additional baselines.**
>
> We thank the reviewer for this suggestion and acknowledge the importance of evaluating our method on popular NLG benchmarks like GSM8k and Humaneval. We have conducted extensive experiments on GSM8k and HumanEval using Llama-3.2-3B-Instruct as the base model for all the PEFT techniques. We have also incorporated three additional baselines -- LoRA+, IA3, and Prompt tuning. For each PEFT method, we report the best run (max) performance, identified through hyperparameter tuning, and the extrinsic robustness score. For GSM8k, we calculate strict match and flexible extract scores. For HumanEval, we calculate pass@2 and pass@4 scores.
>
> Results on GSM8k:
>
> | PEFT | Strict Match(max) | Flexible Extract(max) | Strict match(extrinsic robustness) | Flexible extract(extrinsic robustness) |
> | --- | --- | --- | --- | --- |
> | LoRA | 0.70 | 0.71 | 0.65 | 0.65 |
> | DoRA | 0.75 | 0.75 | 0.67 | 0.67 |
> | LoRA+ | 0.76 | 0.76 | 0.71 | 0.71 |
> | IA3 | 0.76 | 0.77 | 0.74 | 0.75 |
> | Prompt Tuning | 0.71 | 0.71 | 0.68 | 0.68 |
> | MonteCLoRA (ours) | **0.78** | **0.78** | **0.75** | **0.76** |
>
> Results on Humaneval:
>
> | PEFT | pass@2 (max) | pass@4 (max) | pass@2 (extrinsic robustness) | pass@4(extrinsic robustness) |
> | --- | --- | --- | --- | --- |
> | LoRA | 0.52 | 0.58 | 0.50 | 0.55 |
> | DoRA | 0.55 | 0.60 | 0.51 | 0.54 |
> | LoRA+ | 0.56 | 0.60 | 0.53 | 0.58 |
> | IA3 | 0.56 | 0.63 | **0.56** | **0.62** |
> | Prompt Tuning | 0.45 | 0.54 | 0.45 | 0.54 |
> | MonteCLoRA (ours) | **0.58** | **0.65** | 0.55 | 0.61 |
>
> MonteCLoRA achieves top scores on GSM8k with 0.78 strict match and 0.78 flexible extract accuracy, reducing performance spread to 1.26 points, 51% lower than LoRA (2.6) and 47.5% lower than DoRA (2.4). On HumanEval, it secures the highest pass@2 (0.58) and pass@4 (0.65), with robust median scores of 0.55 and 0.61. In comparison, IA3, its strongest baseline achieves a slightly lower pass@2 of 0.54 and pass@4 of 0.60. MonteCLoRA's spread is only 0.4, reducing variability by 89% over LoRA and 63% over DoRA, demonstrating superior robustness and consistency across both mathematical reasoning and code generation tasks.
>
> **R1: Clarification on Motivation**
>
> We have revised Figure 1 with MonteCLoRA. Although the inter-quartile range (IQR) for LoRA remains modest for most tasks, the high distribution spreads suggest that both LoRA and full fine-tuning are sensitive to marginal cases. However, as we discuss in section 3 of our paper, distributional spread is not always the best measure of robustness due to outlier sensitivity (full fine-tuning more sensitive to outliers than LoRA), and distribution median forms a better metric for measuring robustness. We observe that the median performance of full fine-tuning is higher than LoRA in 4 out of 5 tasks. We further observe that the accuracy spread for LoRA, particularly for generative tasks, remains considerably high relative to alternative methods, and median accuracy is also reasonably low compared to other methods, underscoring the need for more stable formulation of low-rank adaption methods for fine-tuning LLMs. MonteCLoRA demonstrates better average spread and median performance than Full FT and LoRA, highlighting the significance of appropriate parameterization for achieving stable low-rank adaptation in LLMs.

---

> ### Author Response · Authors · 2025-06-15
> **Response to Reviewer QNMP Comments - part II**
>
> **W2: The proposed method is 2x slower than LoRA**
>
> The main bottleneck of the proposed method is in the sampling parts, which are typically very slow.
>
> **Parameters**
>
> - $n$ — number of monte-carlo samples
> - $d_{\text{in}}$ — input dimension
> - $d_{\text{out}}$ — output dimension
>
> ---
>
> ### Per-iteration sampling cost
>
> 1. **Multivariate normal sample**
>     - Generate $n$ matrices $Z_{\text{mvn}}\in\mathbb{R}^{d_{\text{in}}\times d_{\text{out}}}$ with i.i.d. $\mathcal{N}(0,1)$ entries.
>     - Cost: $\Theta\bigl(n \cdot d_{\text{in}} \cdot d_{\text{out}}\bigr)$.
> 2. **Wishart sample**
>     - Sample $Z_{\text{wish}}\sim\mathcal{W}(d_{\text{out}}, I)$ via the Bartlett (Cholesky-type) decomposition.
>     - Cost: $\Theta(d_{\text{out}}^{3})$.
> 3. **Dirichlet sample**
>     - Generate $n$ standard-normal scalars (later normalised to obtain $Z_{\text{dir}}$).
>     - Cost: $\Theta(n)$.
>
> ---
>
> ### Total arithmetic cost per generation cycle
>
> $\Theta\bigl(n \cdot d_{\text{in}} \cdot d_{\text{out}} + d_{\text{out}}^{3} + n\bigr) \approx \Theta\bigl(n \cdot d_{\text{in}} \cdot d_{\text{out}} + d_{\text{out}}^{3}\bigr)$ because the linear term $\Theta(n)$ is negligible unless $d_{\text{out}}$ is bound by a very small constant.
>
> This sampling cost constitutes a severe bottleneck when executed sequentially during the model's forward pass. We propose decoupling sample generation from inference by pre-generating samples and storing them in a buffer to address this. During the forward pass, a sample can be retrieved with an O(1) lookup from the buffer, eliminating latency from the critical path.
>
> We implement a fixed-size buffer of samples, which is periodically replenished by a background thread, significantly reducing runtime overhead. Experiments were conducted on the HumanEval dataset using the LLaMA-3-3B-Instruct, LLaMA-3-8B-Instruct, and LLaMA-2-13B models for 1000 training steps, with a training batch size of 8, maximum sequence lengths of 512, and a buffer size of 15. We additionally benchmark the computational cost of MonteCLoRA with the RoBERTa-base model on the CoLA task. The results are summarized below:
>
> | Model                      | Modules | Runtime (× LoRA) | Memory (× LoRA) |
> |---------------------------|---------|------------------|-----------------|
> | RoBERTa-base              | QKV     | 1.63×            | 1.10×           |
> |                           | All     | 1.75×            | 1.15×           |
> | LLaMA-3.2-3B-Instruct     | QKV     | 1.45×            | 1.07×           |
> |                           | All     | 1.78×            | 1.25×           |
> | LLaMA-3.1-8B-Instruct     | QKV     | 1.23×            | 1.06×           |
> |                           | All     | 1.53×            | 1.23×           |
> | LLaMA-2-13B               | QKV     | 1.27×            | 1.09×           |
> |                           | All     | 1.47×            | 1.13×           |
>
> We observe that for larger models, the runtime closely approaches that of LoRA while the memory overhead remains modest. The cost of sampling is effectively amortized across multiple forward passes, and since the sampling can be parallelized on GPU, the wall-clock time for replenishing the buffer is minimal. Furthermore, in larger models, the relative cost of sampling diminishes as their forward pass duration dominates the total step time.

---

> > ### Author Response · Authors · 2025-07-04
> > **Request to check our responses**
> >
> > Dear Reviewer QNMP,
> >
> > We carefully addressed all your concerns. Could you please check our responses and let us know if any other modifications are required.
> >
> > Thanks,
> > Authors

---

> > > ### Comment · Reviewer_QNMP · 2025-07-04
> > > **Satisfied with changes**
> > >
> > > Dear Authors,
> > >
> > > I am satisfied with the inclusion of the other PEFT methods and the additional latency analysis conducted.

---

> > > > ### Author Response · Authors · 2025-07-04
> > > > **Thank you**
> > > >
> > > > Thank you very much.

---

### Review · Reviewer_2dxC · 2025-05-03

**Summary Of Contributions:**

The authors propose a Bayesian-style PEFT method called MonteCLoRA. They propose Bayesian-style priors and sampling for the low-rank parameters used in LoRA and prove various properties of the estimation. Empirical evaluations show similar performance to LoRA and reduced variance in parameter estimates in some cases.

**Audience:**

Yes

**Claims And Evidence:**

No

**Requested Changes:**

- Please address the weaknesses mentioned above.

**Strengths And Weaknesses:**

- The authors recognize the issue of instability in parameter-efficient finetuning methods like LoRA and propose a Bayesian approach to solve the problem.

- The authors provide good analysis of the various properties of the Bayesian estimates and also ablation studies.

- For weaknesses, one main issue is the empirical results do not seem to support the conclusions. For example in Figure 4, LoRA and MonteCLoRA have similar accuracies, but the spread of accuracy is actually larger for MonteCLoRA in 4 out of 5 datasets. The same is true of other tables comparing LoRA and MonteCLoRA like Table 5 and Table 6. The results on Llama2-7B seems to be better in terms of reducing estimation variance, but the accuracy is no better. Also, why do the authors not evaluate on the full GLUE dataset?

- In this approach we still need to choose the hyperparameters for LoRA like rank r and \alpha, learning rate and batch size. In a fully Bayesian approach we should be able to infer from data.

- There are also many clarity issues:
i. In Figure 1: is the instability shown due to choice of different random seeds or different hyperparameters? It is not detailed there.
ii. Algorithm 1 is not clear. What does the last step of updating the weight matrix mean? Is it done by SGD? And shouldn't we be updating the trainable parameters of the different distributions?
iii. The scale factor \epsilon in Lemma 4.1 does not seem to be defined anywhere and only appears once in Algorithm 1 but not explained. What is it?

---

> ### Author Response · Authors · 2025-05-18
> **Acknowledgement of reviewer 2dxC comments**
>
> Dear reviwerer 2dxC,
>
> We appreciate your valuable comments on our paper. We are working on addressing your comments and will be providing the clarifications soon.
>
> Thanks,

---

> > ### Author Response · Authors · 2025-06-22
> > **Response to Reviewer 2dxC Comments - Part II**
> >
> > ### W2: Hyperparameter selection with MonteCLoRA
> >
> > We thank the reviewer for pointing out the need for hyperparameter selection in our approach. However, we would like to clarify a potential misunderstanding. The **primary contribution** of our work is not to eliminate the use of hyperparameters altogether, but rather to **make fine-tuning significantly more robust to their choice**.
> >
> > While a fully Bayesian treatment could, in principle, marginalize over architectural hyperparameters like the rank $r$, this is orthogonal to the objective of our work. Our aim is to **retain the practical benefits of LoRA’s parameter efficiency** while **introducing Bayesian reparameterisation to reduce estimator variance, mitigate mode collapse, and enhance convergence stability**, as detailed in Sections 3 and 4 of our paper.
> >
> > Empirically, MonteCLoRA demonstrates superior robustness (e.g., 62% lower spread and 13.3% higher robustness than LoRA on LLaMA-3.2 tasks) **without altering any task-specific hyperparameter grids**, suggesting its utility as a plug-in replacement for LoRA in real-world pipelines.
> >
> > We have used $\text{rank} = 8, \alpha = 8$ for our main experiments because this was used in the original LoRA paper. Keeping the other hyperparameters same and keeping $\alpha = 16$ resulted in suboptimal performance for LoRA, which is also why we leaned towards using $\alpha = 8$. But the more important thing to note here was that **even though performance degraded very significantly for LoRA at** $\text{learning rate} = 1\times10^{-3}$ **(the highest learning rate considered), MonteCLoRA stayed consistent, with high accuracy. LoRA diverged very quickly.**
> >
> > The maximum accuracies are reported below for LoRA and MonteCLoRA at this learning rate:
> >
> > | TASK | Maximum Accuracy (LoRA) at $\texttt{lr = 1e-3}$ | Maximum Accuracy (MonteCLoRA) at $\texttt{lr = 1e-3}$ |
> > | --- | --- | --- |
> > | SST2 | 0.9186 | 0.9392 |
> > | QNLI | 0.5054 | 0.9174 |
> > | QQP | 0.6318 | 0.8792 |
> > | MNLI | 0.3544 | 0.8240 |
> > | Average | 0.6026 | **0.8900** |
> >
> > For SST2, the maximum accuracy is nearer to MonteCLoRA’s, but it still diverged after 3 epochs, and accuracy plummeted to $\sim$51% at the end of 10 epochs. In contrast, MonteCLoRA stably converged to an expected level of accuracy.
> >
> > ### W3: Clarity issues
> >
> > 1. As mentioned in the explanation for figure 1, the instability is due to choice of different hyperparameters, such as learning rates and training batch sizes.
> >
> > 2. Algorithm 1 is now changed in the manuscript, with the intention to provide more clarity to the reviewers. The following should address the concerns:
> >
> >            2.a. In the last step we are replacing the current weight matrix with the MonteCLoRA estimation that we have done. The loss and gradients are calculated with this matrix.
> >
> >            2.b. The distributions are trainable, as we have used reparameterised sampling and are trained during back-propagation along with the rest of the model.
> >
> > 3. The scale factor was earlier missing from the algorithm. We have added it and the use and effect of sample scaler is now defined in section 4.3.

---

> ### Author Response · Authors · 2025-06-22
> **Response to Reviewer 2dxC Comments - Part I**
>
> We thank reviewer 2dxC for their valuable comments. We address the raised queries in the following responses. We have also made the necessary changes in our revised manuscript (highlighted in magenta font).
>
> ### W1.a Results on GLUE large tasks
>
> We thank the reviewer for suggesting the additional experiments. We have added the results with full fine-tuning (FFT), LoRA, AdaLoRA and MonteCLoRA on SST-2, MNLI, QNLI and QQP tasks.
>
> | Method | Metric | MRPC | CoLA | RTE | WiC | BoolQ | SST2 | QNLI | QQP | MNLI | Average |
> | --- | --- | --- | --- | --- | --- | --- | --- | --- | --- | --- | --- |
> | Full FT |  | 89.2 | 84.0 | 76.5 | 70.4 | 80.4 | **94.6** | **92.2** | **88.4** | **85.6** | **84.6** |
> | LoRA | Accuracy ↑ | 89.9 | **85.1** | 80.1 | 67.5 | 80.0 | 93.1 | 89.1 | 87.0 | 83.0 | 83.9 |
> | AdaLoRA |  | 88.7 | 84.5 | **81.2** | 67.7 | **80.6** | 92.4 | 89.7 | 86.6 | 84.2 | 84.0 |
> | MonteCLoRA |  | **91.2** | 84.9 | **81.2** | **71.3** | 79.9 | 92.4 | 89.8 | 85.7 | 83.5 | 84.4 |
> |  |  |  |  |  |  |  |  |  |  |  |  |
> | Full FT |  | 0.34 | **0.40** | 0.60 | 0.68 | 0.51 | 0.26 | 0.28 | **0.28** | **0.39** | 0.42 |
> | LoRA | NLL ↓ | 0.32 | **0.40** | 0.54 | 0.62 | **0.49** | **0.20** | 0.27 | 0.31 | 0.44 | 0.40 |
> | AdaLoRA |  | 0.42 | 0.54 | 0.62 | 0.83 | 0.55 | 0.21 | 0.28 | 0.30 | 0.42 | 0.46 |
> | MonteCLoRA |  | **0.31** | 0.41 | **0.46** | **0.60** | 0.50 | 0.21 | **0.26** | 0.32 | 0.44 | **0.39** |
>
> The robustness scores are highlighted below
>
> | Method | Robustness | MRPC | CoLA | RTE | WiC | BoolQ | SST2 | QNLI | QQP | MNLI | Average |
> | --- | --- | --- | --- | --- | --- | --- | --- | --- | --- | --- | --- |
> | Full FT |  | 2.21 | 1.98 | 1.45 | 1.45 | 1.71 | 1.91 | 2.06 | 2.05 | 1.32 | 1.79 |
> | LoRA | Intrinsic ↑ | 3.11 | 2.38 | 1.86 | 1.49 | **1.76** | **4.81** | 2.87 | 3.06 | 2.11 | **2.61** |
> | AdaLoRA |  | 1.48 | 1.61 | 0.97 | 0.55 | 0.93 | 3.79 | 2.91 | **3.12** | 2.14 | 1.94 |
> | MonteCLoRA |  | **3.16** | **2.43** | **2.18** | **1.63** | 1.75 | 4.09 | **3.20** | 2.81 | **2.27** | **2.61** |
> |  |  |  |  |  |  |  |  |  |  |  |  |
> | Full FT |  | 89.1 | 83.8 | 52.7 | 63.0 | 80.1 | 72.7 | 84.0 | 81.4 | 60.3 | 74.1 |
> | LoRA | Extrinsic ↑ | 88.6 | 83.8 | 80.1 | 60.6 | 80.0 | **92.6** | 85.3 | 85.5 | 81.7 | 82.0 |
> | AdaLoRA |  | 87.7 | 82.6 | 78.5 | 65.7 | **80.2** | 90.0 | 86.0 | **85.8** | 82.1 | 82.1 |
> | MonteCLoRA |  | **89.3** | **84.3** | **81.2** | **69.2** | 79.9 | 91.0 | **89.6** | 85.0 | **83.3** | **83.6** |
> |  |  |  |  |  |  |  |  |  |  |  |  |
>
> On average, MonteCLoRA performs 1.5% better robustness and 0.4% better accuracy than the best baselines, AdaLoRA.
>
> ### W1.b LLaMA-7B results
>
> We thank the reviewer for their observation regarding the LLaMA-7B results. While it is correct that the **average accuracy remains comparable to LoRA**, the key insight from these results lies in the **significant reduction in estimation variance**, reflected in the much **lower spread** and similar **extrinsic robustness** scores. This suggests that MonteCLoRA provides **more consistent and stable fine-tuning outcomes**, which is particularly valuable in practical deployments where reproducibility and stability across runs are critical.
>
> We acknowledge that spread/variance alone is not a comprehensive metric of robustness. However, the fact that MonteCLoRA maintains performance parity with LoRA while **substantially reducing variance** highlights its strength in **stabilising the optimisation landscape** during fine-tuning.
>
> Moreover, we would like to draw attention to our results on the **LLaMA-3.2-3B-Instruct** model, evaluated on **GSM8k** and **HumanEval**. Here, MonteCLoRA demonstrates **consistent improvements over LoRA and other baselines**, achieving gains in **both accuracy and robustness metrics**.
>
> Taken together, these results underscore that MonteCLoRA not only matches or exceeds LoRA in robustness and accuracy but also introduces a **principled Bayesian mechanism** to improve consistency across tasks and architectures -  especially evident in newer, more challenging model settings.

---

> ### Author Response · Authors · 2025-07-04
> **Request to check our responses**
>
> Dear Reviewer 2dxC,
>
> We carefully addressed all your concerns. Could you please check our responses and let us know if any other modifications are required.
>
> Thanks
>
> Authors

---

> > ### Author Response · Authors · 2025-07-10
> > **Reminder to review our responses**
> >
> > Dear Reviewer 2dxC,
> >
> > Thank you for your valuable comments.
> >
> > We would like to note that two other reviewers have acknowledged our responses and expressed satisfaction.
> >
> > Could you kindly review our responses to your comments and let us know if any further clarifications are needed?
> >
> > Thanks
> > Authors

---

> > > ### Comment · Reviewer_2dxC · 2025-07-10
> > >
> > > Thank you for your updates, they greatly improve the paper.

---

> > > > ### Author Response · Authors · 2025-07-10
> > > >
> > > > Thank you very much.

---

### Review · Reviewer_rGiY · 2025-06-20

**Summary Of Contributions:**

The paper introduces MonteCLORA, a Bayesian approach to improve the efficiency and robustness of fine-tuning large language models (LLMs) using low-rank adaptation. Its key contributions are:
- **Theoretical Analysis**: It explores how hyperparameters influence LLM fine-tuning, focusing on the sensitivity of low-rank methods like LoRA to learning rates and batch sizes. Mathematical proofs tie these choices to optimization stability.
- **Proposed Method**: MonteCLORA leverages Monte Carlo estimation to learn posterior distributions of low-rank parameters, modeled as mixtures of multivariate Gaussian distributions with Wishart priors. This boosts stability while adding just O(r) extra parameters.
- **Empirical Validation**: The method was tested on five natural language understanding tasks and six generation tasks with models like RoBERTa-base and LLaMA-1-7B. It achieved up to 3.8% better accuracy, 8.6% more robustness, and a 62% smaller performance spread with LLaMA-3.2-3B-Instruct.

**Audience:**

Yes

**Claims And Evidence:**

Yes

**Requested Changes:**

1. **Quantify Computational Trade-offs**: Add a clear breakdown of how Monte Carlo sampling increases training time—think benchmarks like hours per epoch—and highlight cases where the gains outweigh the costs.
2. **Outline Limitations**: Add a subsection to discuss limitations of this work.
3. **Outline Future Directions**: Suggest next steps, like making sampling more efficient, trying MonteCLORA in pre-training, or scaling it to bigger models.
4. **Figure 1 Fix**: Move the numbers above or below the boxes—right now, they overlap with lines, making it hard to read.
5. **Figure 3 Caption**: Add a quick summary of each step in the caption so readers can follow the process without flipping back to the text.
6. **Lemma 3.1 Clarification**: Define \(n\) upfront—it’s not clear what it stands for yet.
7. **Lemma 3.2 Notation**: Shift the tilde to \(\tilde{J}(\theta)\) for clearer notation.
8. **Lemma 3.3 Reference**: Fix the reference to “Lemma 3”—it’s probably meant to be Lemma 3.2?
9.  **Lemma 4.1 Assumption**: Spell out the assumption behind the norm (e.g., is \(X \geq 0\) required?) to make the step airtight.
10. **Lemma 4.5 Detail**: Explain where \(\sum = 1\) comes from—it’s not obvious without more context.
11. **Consistent Color Coding**: Stick to the same colors for each method across all figures (e.g., MonteCLORA in red) to make them easier to track.
12. **Comparison Tables/Figures**: Add arrows (↑/↓) in all tables and all figures to show whether higher or lower values are better, like in Table 5, for quicker understanding.

**Strengths And Weaknesses:**

## Strengths
- **Innovative Approach**: MonteCLORA blends Bayesian inference with Monte Carlo methods, offering a new way to tackle hyperparameter sensitivity in fine-tuning.
- **Solid Theoretical Grounding**: Detailed mathematical analysis backs up its claims about stability, giving it a strong foundation.
- **Strong Empirical Results**: Across various tasks and models, it consistently improves accuracy, robustness, and convergence—like a 53% lower performance spread in NLG tasks with LLaMA-1-7B.

## Weaknesses
- **Increased Computational Cost**: Monte Carlo sampling pushes training time complexity to O(n_in r² + N), which might hinder scalability for big models or limited resources.
- **Implementation Complexity**: The Bayesian setup could be tricky for users who aren’t familiar with probabilistic methods, even with the theory explained.
- **New Hyperparameters**: MonteCLORA brings in its own parameters, like the number of Monte Carlo samples, which still need some tweaking.
- **Limited Generalizability Testing**: While effective on tested models and tasks, the method lacks evaluation across a broader range of LLM architectures or domains (e.g., beyond HumanEval for code generation).
  - *Update*: Given the inclusion of multiple additional experiments, this concern appears to be sufficiently addressed. Consider removing this point to reflect the updated scope of the evaluation.

---

> ### Author Response · Authors · 2025-06-24
> **Response to Reviewer rGiY Comments**
>
> We thank reviewer rGiY for their valuable comments. We address the raised queries in the following responses. We have also made the necessary changes in our revised manuscript (highlighted in orange font).
>
> **W1:  Increased Computational Cost**
>
> To address this issue, as also pointed out earlier by reviewer QNMP, we implement a fixed-sized buffer of samples, periodically replenished by a background thread, significantly reducing runtime overhead. In Table 12 of the updated manuscript, we see that for larger models, the runtime of MonteCLoRA closely approaches that of LoRA, showing that there should not be any hindrance in scalability for big models or limited resources.
>
> **W2: Implementation Complexity**
>
> We appreciate the reviewer’s concern regarding the implementation complexity of a Bayesian formulation. While MonteCLoRA does involve sampling from multiple distributions and manipulating low-rank weight matrices, we have taken significant steps to make the method accessible to practitioners. Specifically, we have integrated MonteCLoRA into the popular Huggingface PEFT library as a seamless extension of the LoRA framework. Switching from LoRA to MonteCLoRA requires minimal changes - users only need to set a few additional flags, with no need to directly handle the probabilistic machinery.
>
> Moreover, our implementation is compatible with the latest versions of the Hugging Face transformers library and supports distributed training out of the box. We have thoroughly tested the codebase across diverse GPU architectures, including A100, A6000, and V100, ensuring consistent behavior and reproducibility across hardware. We believe this level of integration and validation significantly lowers the barrier to adoption and makes MonteCLoRA practical for real-world use.
>
> **W3: New Hyperparameters**
>
> We agree that MonteCLoRA introduces a few new hyperparameters, such as the number of Monte Carlo samples and the sample scaler $\epsilon$. However, we would like to emphasize that these parameters are *not highly sensitive* and were fixed across all tasks and model scales in our experiments without per-task tuning. We used the same default configuration (e.g., $N=4$, $\epsilon=5 \times 10^{-3}$) for both NLU and NLG tasks (except LLaMA-3.2 where we used $\epsilon=2.5 \times 10^{-5}$ and observed consistently strong performance and stability.
>
> Moreover, our theoretical formulation shows that these hyperparameters primarily influence the variance of the posterior estimate, rather than the convergence behavior. As such, they are more robust and easier to tune than conventional hyperparameters like learning rate or batch size, which tend to have a much larger effect on downstream performance.
>
> We have also added a theoretical analysis explaining the role of $\epsilon$ and how it scales the model's output. Based on this, we provide a simple practical guideline: as model size increases, $\epsilon$ should be proportionally decreased. These insights, along with suggested defaults, have been included in the manuscript to help practitioners adopt MonteCLoRA without the need for extensive hyperparameter tuning.
>
> **RC1**
>
> Thank you for pointing this out. We have addressed this by including a detailed theoretical breakdown of the computational complexity introduced by the Monte Carlo sampling procedure in Appendix C. To complement this, we now provide empirical benchmarks comparing MonteCLoRA and LoRA in terms of actual training time. Specifically, we report runtime metrics such as hours per epoch across different model scales.
>
> Our experiments show that for larger models (e.g., LLaMA-3.1-8B, LLaMA-2-13B), MonteCLoRA incurs only a modest 1.2–1.3× increase in training time per epoch compared to LoRA. This overhead remains stable across hardware types and is significantly offset by the improvements in stability, robustness, and downstream task performance reported in the main paper. We believe this trade-off is favourable, especially in settings where fine-tuning stability is critical.
>
> **RC2**
>
> The Limitations section is now updated in the manuscript.
>
> **RC3**
>
> The outline future direction sections is now included in the manuscript.
>
> **RC4**
>
> Figure 1 has now been changed in the manuscript, and the numbers have also been placed at the bottom of bars for better readability.
>
> **RC5**
>
> Figure 3 caption has been updated in the manuscript, providing more detail for the ease of the readers to follow the process.
>
> **RC6 - RC10**
>
> Changes added to the main text.
>
> **RC11**
>
> Updated figures in the main text.
>
> **RC12**
>
> Tables for GLUE already had arrows. Added arrows in column headers of generative tasks.

---

> > ### Comment · Reviewer_rGiY · 2025-06-26
> > **Response to Authors**
> >
> > Thank you for your response and for addressing my comments.
> >
> > I have a minor suggestion regarding the structure of your manuscript: I recommend placing the "Limitations" and "Future Work" content as subsections within the "Discussion" section. This is a common and often preferred organizational structure in many academic journals.

---

> ### Author Response · Authors · 2025-06-26
> **Response to Reviewer rGiY Comments**
>
> We thank Reviewer rGiY for their suggestion regarding the structure of the manuscript.
>
> We have made the suggested changes in our revised manuscript (highlighted in teal)

---

### Author Response · Authors · 2025-06-24
**Rebuttal Summary from Authors**

Dear EiC, AE, and reviewer rGiY, 2dxC, and QNMP,

This is to confirm that we have addressed all of your comments. Below is a summary of our responses.

**Computational complexity of MonteCLoRA**

We have implemented a fixed-sized buffer of samples, periodically replenished by a background thread, significantly reducing runtime overhead. In Table 12 of the updated manuscript, we see that for larger models, the runtime of MonteCLoRA closely approaches that of LoRA, showing that there should not be any hindrance in scalability for big models or limited resources. Our experiments show that for larger models (e.g., LLaMA-3.1-8B, LLaMA-2-13B), MonteCLoRA incurs only a modest 1.2–1.3× increase in training time per epoch compared to LoRA. This overhead remains stable across hardware types and is significantly offset by the improvements in stability, robustness, and downstream task performance reported in the main paper. We believe this trade-off is favourable, especially in settings where fine-tuning stability is critical.

**Experimental results of MonteCLoRA on GLUE large tasks**

We have added the results for all the PEFT baselines (LoRA, AdaLoRA, and MonteCLoRA, along with full fine-tuning) with RoBERTa-base on large GLUE tasks (SST-2, MNLI, QNLI, QQP). MonteCLoRA achieves 0.9 and 12.6 points better extrinsic robustness than LoRA and full fine-tuning, respectively. MonteCLoRA shows the least accuracy spread (5.4 points), compared to LoRA (19.4) and full fine-tuning (28.7), indicating much more stable performance across hyperparameter settings.

**Experimental results of MonteCLoRA on HumanEval and GSM8k datasets**

We have conducted extensive experiments with LLaMA-3.2-3B on GSM8k and HumanEval datasets. On GSM8k, MonteCLoRA achieves the highest strict match (0.78) and flexible extract (0.78) scores, 1.5\% better than IA3 and LoRA+, the best baselines (we added 3 new baselines -- IA3, LoRA+, and Prompt tuning). On HumanEval, MonteCLoRA achieves the highest pass@2 (0.58) and pass@4 (0.65) scores. In conclusion, while traditional methods such as LoRA, LoRA+, and IA3 provide competitive baselines, MonteCLoRA balances top-tier accuracy and robustness, solidifying its effectiveness for fine-tuning complex NLG tasks involving symbolic reasoning and code generation.

We have revised our manuscript to incorporate these changes (color blue for reviewer QNMP, magenta for reviewer 2dxC, orange for reviewer rGiY). We hope these responses address all your concerns. We are looking forward to your responses and any further queries you may have.

---

### Decision · Action_Editor_zpTV · 2025-07-27

**Recommendation:** Accept as is

**Additional Comments:**

All 3 reviewers recommended to accept the manuscript which was improved during the rebuttal and discussion.

**Audience:**

Yes

**Audience Explanation:**

The topic of low rank adapter and fine tuning is quite relevant to TMLR's audience and the submission provides an interesting contribution on the topic.

**Claims And Evidence:**

Yes

**Claims Explanation:**

The claims are well supported by theoretical and experimental results.